# Evapotranspiration prediction for European forest sites does not improve with assimilation of in-situ soil water content data

Lukas Strebel[1,2,3], Heye Bogena[1,2] Harry Vereecken[1,2], Mie Andreasen[4], Sergio Aranda-Barranco[5], and Harrie-Jan Hendricks Franssen[1,2]

[1] Agrosphere Institute, IBG-3, Forschungszentrum Jülich GmbH, Germany

[2] Centre for High-Performance Scientific Computing in Terrestrial Systems: HPSC TerrSys, Geoverbund ABC/J, Leo-Brandt-Strasse, 52425 Jülich, Germany

[3] Institute for Applied Geophysics and Geothermal Energy, RWTH Aachen University, Aachen, Germany

[4] Geological Survey of Denmark and Greenland, Copenhagen, Denmark

10    [5] Department of Ecology, University of Granada, Granada 18071, Spain

*Correspondence to*: Lukas Strebel (l.strebel@fz-juelich.de)

**Abstract.** Land surface models (LSM) are an important tool for advancing our knowledge of the Earth system. LSM are constantly improved to represent the various terrestrial processes in more detail. High quality data, freely available from various observation networks, are providing being used to improve the prediction of terrestrial states and fluxes of water and energy. To optimize LSM with observations, data assimilation methods and tools have been developed in the past decades. We apply the coupled Community Land Model version 5 (CLM5) and Parallel Data Assimilation Framework (PDAF) system (CLM5-PDAF) for thirteen forest field sites throughout Europe covering different climate zones. The goal of this study is to assimilate in-situ soil moisture measurements into CLM5 to improve the modeled evapotranspiration fluxes. The modeled fluxes will be evaluated using the predicted evapotranspiration fluxes with eddy covariance (EC) systems. Most of the sites use point scale measurements from, however for three of the forest sites we use soil water content data from cosmic-ray neutron sensors, which have a measurement scale closer to the typical land surface model grid scale and EC footprint. Our results show that while data assimilation reduced the root-mean-square error for soil water content on average by 56 to 64%, the root-mean-square error for the evapotranspiration estimation is increased by 4%. This finding indicates that only improving the SWC estimation of state of the art LSM such as CLM5 is not sufficient to improve evapotranspiration estimates for forest sites. To improve evapotranspiration estimates, it is

also necessary to consider the representation of LAI in magnitude and timing, as well as uncertainties in water uptake by roots and vegetation parameters.

## 1 Introduction

Land surface models (LSM) are important tools to improve our understanding of the Earth system. LSM cover a broad range of land surface processes like the partitioning of incoming energy at the land surface, mass exchange between the land and atmosphere, hydrological and ecological processes. They use sophisticated parameterizations and are constantly improved to achieve a more accurate representation of land surface processes, e.g. Arora et al. (2020) and references therein. However, there are still many sources of uncertainty introducing systematic biases in the LSM (e.g. initial conditions, atmospheric forcings, parameters, and parameterization). One approach to improve model predictions is to assimilate observational data. Improved estimates of evapotranspiration (ET) by LSM are of main interest as ET is a major driver of climate and weather, an important component of the water and energy cycles, closely coupled to the carbon cycle through the photosynthesis process (Jung et al., 2011). Fine spatial scale ET estimations are important to estimate water use and plant stress (Wurster et al. 2020). The flux of ET is, however, influenced by multiple factors, including soil water content (SWC), soil properties, ecophysiological processes, and vegetation characteristics (Wilson et al., 2004), so it is more common to assimilate these prognostic variables rather than ET itself.

Many studies assimilate soil moisture products into LSMs (e.g. Hung et al., 2022; Mahmood et al., 2019; Naz et al., 2019; Liu and Mishra., 2017; Han et al., 2015) and report on the impact on hydrological variables like root-zone-moisture and runoff. Some studies use assimilation of soil water content or related variables to evaluate ET estimation of LSMs. For example, Girotto et al. (2017) assimilated terrestrial water storage from the Gravity Recovery and Climate Experiment into a land surface model and evaluated results over India. They found that the assimilation decreased the accuracy of ET estimation compared to observations due to model limitations in representing irrigation. Peters-Lidard et al. (2011) assimilated two different remotely sensed soil water content products into the Noah land surface model over North America and found mixed results regarding the improvement of latent heat flux estimates. The domain averaged root-mean-square error of the latent heat flux reduced from 27.6 Wm$^{-2}$ to 25.6 Wm$^{-2}$ or increased to 29.4 Wm$^{-2}$ depending on the assimilated soil water content product. Additionally, they show that the improvements and degradation vary spatially across their study domain, with land cover type, and as function of the season and they note that the most significant improvements occur for cropland and grassland. Liu and Mishra (2017) assimilated surface soil water content data from the Advanced Microwave Scanning Radiometer-Earth Observing System in a global Community Land Model version

4.5 and found ET bias reductions of up to 2.5mm/day compared to the Global Land Data Assimilation System (GLDAS) data product.

For our study, we chose the latest version (version 5) of the widely used Community Land Model (CLM5) (Lawrence et al. 2019) as various land surface process representations have been improved in CLM5 compared to earlier versions. For instance, Kennedy et al. (2019) added a plant hydraulic stress parameterization to improve the accuracy of simulated transpiration and soil water content. Lawrence et al. (2019) demonstrated the improvements of CLM5 over its precursor CLM4 in terms of ET using two study sites as examples and highlighted the better representation of the effects of soil depth on ET prediction in CLM5. On the other hand, Cheng et al. (2021) found that CLM5 predicts lower ET compared to older CLM versions and various observational data, likely due to low photosynthetic rate and leaf area index (LAI), which is consistent with their finding of low gross primary production (GPP) compared to reference data in the same simulations. In addition to these regional to global validation studies, CLM was used in several single point setups, i.e. simulations for a single grid cell, to evaluate the performance of various LSM components. For example, Hudiburg et al. (2013) used CLM 4.0 to estimate net primary production (NPP) and GPP of a forested site and compared it with eddy covariance (EC) measurements. Another study (Zhang et al., 2019) reduced an overestimation of growing-season LAI and annual GPP of a grassland site for a CLM 4.5 single-point setup. More recently, CLM5 was extended to consider both cover crop management with improvements to ET estimation of up to 57% (Boas et al., 2021) and fruit tree cultivation using extensive field measurements with high correlation between observed and modeled ET (Dombrowski et al., 2022. Other studies have used manual tuning of parameters to improve CLM simulations for forests. For instance, Duarte et al. (2017) calibrated CLM4.5 for an old-growth coniferous forest and found good agreement between simulated and observed response of canopy conductance to atmospheric vapor pressure deficit and soil water content. Raczka et al. (2016) used CLM4.5 and implemented a seasonally varying calibration of vegetation parameters and accurately simulated net carbon exchange, latent heat exchange, and biomass.

In this study, we investigate if assimilating high quality, in-situ soil water content measurements can improve the evapotranspiration estimates of LSM. We focus on one specific land cover type, namely forests. In a previous study (Strebel et al. 2022), we investigated the potential for data assimilation of in-situ SWC measurements to improve model estimation for a single forest site. This study expands this method to more forest sites and investigates the effect of improved SWC estimation on ET. Investigating the method for a large number of sites is the important contribution of this study and necessary to show that the conclusions from Strebel et al. (2022) are not just a characteristic of the one study site but apply more broadly to forest sites simulated with CLM5. To investigate this, we use point and plot scale in-situ soil water content measurements. For most sites we use point measurements provided by FLUXNET (Baldocchi et al., 2020) and eLTER Europe. The FLUXNET data have been used in various studies to

verify or compare model results. For example, Dirmeyer et al. (2018) used FLUXNET data to compare four model systems, including CLM4.5, in three configurations and found for annual averaged ET that correlations range from 0.28 to 0.43 and for sensible heat from 0.14 to 0.54. The point scale measurements use invasive equipment and the specific measurement volume, exact depth of the sensors, number of sensors, and number of stations varies from site to site. For a few sites we use soil water content measurements from Cosmic Ray Neutron Sensing (CRNS) from the COSMOS-Europe data set (Bogena et al., 2022). The CRNS provides continuous and non-invasive soil water content measurements over a spatial footprint of hundreds-of-meters and integrates from the surface to a depth of 10-70 cm vertically in the soil (Zreda et al., 2008; Köhli et al., 2015). CRNS use neutrons as proxy for SWC and the vertical measurement depth varies with the soil moisture conditions. Additionally, the uncertainty of CRNS-derived soil moisture varies not only with the different neutron detectors but also with the number of counts in a time period and therefore results under lower soil moisture conditions are more accurate (Bogena et al., 2022). The spatial footprint area is similar to the footprint of the EC flux tower. We use the final processed data on soil water content and vertical penetration (measurement) depth provided by the COSMOS-Europe dataset (Bogena and Ney, 2021). In this study, we use the ensemble Kalman filter to assimilate in-situ soil water content measurements into CLM5 simulations and the effect on the modeling results are quantified by comparing the modeled ET against the observed ET obtained from EC flux towers. We also analyze the effects on other land-atmosphere exchange fluxes, i.e. net ecosystem exchange (NEE) and gross primary production (GPP). The paper is structured as follows: First, we introduce the model and data assimilation framework used. The sites selected for this study and the observational data used for data assimilation and model-observation comparison are then described. Subsequently, the results for each variable of interest are shown and analyzed. Finally, we end with a discussion of the obtained results and conclusions.

## 2. Methods and Materials

### 2.1 Study sites

In our study, we are interested in the characterization of water, energy and carbon exchange between (European) forest ecosystems and the atmosphere, and whether soil water content assimilation can improve the characterization of these processes. Therefore, we selected European sites with different forest types (see Table 1) covering different climate zones in Europe. Another important constraint was the availability of soil water content data and evapotranspiration measurements for the period from 2009 to 2018. The selected sites are mostly part of FLUXNET (Baldocchi et al., 2020) or the European Long-

Term Ecological Research network (eLTER-Europe) (Parr et al., 2002). In addition to the sites from these observation networks, we included three sites from the COSMOS-Europe network (Bogena et al., 2022) where cosmic-ray neutron sensors (CRNSs) are installed to estimate the soil water content of the forested sites. Table 1 gives an overview of all selected sites for this study and Figure 1 shows the distribution on the map.

In this study, daily average soil water content data are assimilated (see section 2.4.1 for more details) and the model is verified using daily average evapotranspiration and sensible heat flux data. Since the observational data were already quality controlled by the providers, we did not filter out any data. We only assimilated (daily mean averaged) soil water content observations when measurements were available for a given day. The daily mean averages were calculated independent from the observation frequency for the different sites. Similarly, simulated evapotranspiration was only compared with observations when data were available, on the basis of daily mean averages.

## 2.2 Model description

For our study, we used the Community Land Model version 5.0 (CLM5) that can be applied in various configurations (Lawrence et al. 2019). We use CLM5-BGC, i.e. CLM5 with the biogeochemistry module active as opposed to CLM5 with fixed phenology. The biogeochemistry module enables a fully prognostic treatment for carbon and nitrogen in the land surface model and has a significant impact on the modeled water and energy budgets.

CLM5 uses a sub-grid hierarchy of various Plant Functional Types (PFTs) to characterize the land-use and vegetation type within every grid cell, e.g. evergreen needle leaf or deciduous broad leaf forests. CLM5 contains a spatially variable soil depth with an underlying, impermeable bedrock instead of the unconfined aquifer parameterization used in the former CLM4 versions. To estimate the soil water content, CLM5 solves the Richard's equation using the Brooks-Corey parameters derived from pedotransfer functions from Clapp and Hornberger (1978) with a finite-difference approximation to represent the vertical discretization and temporal evolution of soil water content.
The sensible and latent heat flux estimation in CLM5 is derived from the Monin-Obukhov Similarity Theory and differentiated for vegetated and non-vegetated surfaces. CLM5 simulates sensible and latent heat flux for both vegetated and ground fluxes. For the vegetation part the contributions from the leaf boundary layer, the sunlit and shaded stomatal resistances affect the total resistance to the modeled water vapor transfer. The water vapor transfer includes transpiration from dry leaf surfaces and the transpiration removes water from the soil based on root fraction for a given soil layer. Interception, throughfall and canopy drip are explicitly modeled in CLM5 and canopy evaporation is represented as from the sum of stem and leaf surface evaporation as a function of temperature. The ground fluxes, e.g. from bare soils or soil beneath a canopy, are dependent on the ground surface temperature. The ground latent heat flux is

reduced if not enough soil moisture is available and the excess energy is redistributed to the sensible heat flux. The detailed procedure and equations are documented in Lawrence et al. (2018).

## 2.3 Data assimilation

### 2.3.1 Ensemble Kalman filter

In this work, assimilation of soil water content measurements is performed with the Ensemble Kalman Filter (EnKF) (Evensen, 1994; Burgers et al., 1998). The EnKF uses an ensemble modeling approach, with various simultaneous model runs, to approximate the model uncertainty. The ensemble members have different input model parameters and atmospheric forcings (see section 2.4 for details). We define a state vector x and an observation vector y, e.g.

$$x^i = \begin{pmatrix} \theta^i_{1,1} \\ \theta^i_{1,2} \\ \dots \\ \theta^i_{1,m} \\ \dots \\ \theta^i_{n,m} \end{pmatrix} (1)$$

where n is the number of layers and m is the number of grid cells, $\theta^i_{j,l}$ is the soil water content for layer j and grid cell l of the model and the superscript i refers to ensemble member i. In this study we use an ensemble of 96 member to sample the model uncertainty.

$$y = o + e \ (2)$$

where o is a vector of the observational data and e represents a perturbation vector with mean zero and covariance according to the observational error covariance matrix. This perturbation vector is used to correct the error statistics as described in Burgers et al. (1998).

The update step of the ensemble Kalman filter is:

$$x^i_a = x^i_f + K\left[y - Hx^i_f\right] (3)$$

where the superscript i refers to ensemble member i, $x^i_a$ is the updated state vector after the analysis, $x^i_f$ is the forecasted model state vector, K is the Kalman gain and H is the measurement operator that transforms between model and observational states. In this study, the measurement operator H consists of a simple mapping of the observations to the corresponding model layers in the state vector for simulations with point measurements. For FLUXNET sites, measured soil water content is provided for up to three depths described as superficial, medium, and deep. Since data assimilation in CLM5-PDAF requires a specific vertical layer, we assigned 5, 20 and 50 cm to the respective FLUXNET SWC layers. For the CRNS sites, the measurement depth for each individual measurement is calculated following

Schrön et al. (2017) and is included in the dataset from Bogena et al. (2022). For simulations assimilating CRNS, H assigns the mean observed SWC to all the layers down to the measurement depth. This is a simplified approach and will be improved in further studies to take the weighting function from Schrön et al. (2017) into account. The Kalman gain is calculated accordingly:

$$K = PH^T(R + HPH^T)^{-1} \ (4)$$

where the superscript T is used for transposed matrices, R is the observational error covariance matrix, and P is the model error covariance matrix, which is approximated through ensemble statistics, specifically:

$$P = \frac{1}{(N-1)}\sum_{i=1}^{N}\left(x_f^i - \overline{x_f}\right)\left(x_f^i - \overline{x_f}\right)^T \ (5)$$

where N is the number of ensemble members and $\bar{x}$ is the ensemble mean.

In this study, the state vector depends on the simulation scenario (explained in more detail in section 2.3.2) and R is based on the measurement errors which are assumed to be constant and independent with a root-mean square error of 0.02 cm³/cm³.

To enable data assimilation with CLM5, we use the Parallel Data Assimilation framework (PDAF) (Nerger et al. 2005), which was recently coupled to CLM5 (Strebel et al. 2022). This coupling (CLM5-PDAF) also supports the assimilation of soil water content measurements.

### 2.3.2 Parameter updating

In addition to the use of data assimilation for state updating, we also perform parameter updating based on the state augmentation approach (Friedland, 1969; Fertig et al., 2009). Here, model parameters are attached to the state vector and updated based on the Kalman gain calculations without observations of the model parameters. By default, CLM5-PDAF updates soil hydraulic parameters through changes to fractions of sand, clay, and organic matter and the pedotransfer function of Clapp and Hornberger (1978). In this indirect approach the state vector for the EnKF is defined as follows:

$$x^i = \begin{pmatrix} \theta^i \\ \%sand^i \\ \%clay^i \\ \%organic^i \end{pmatrix} \ (6)$$

where the superscript i refers to ensemble member i. The components $\theta$, %sand, %clay, and %organic each represent a vector containing the respective variable for each soil layer of each grid cell of the model. A damping factor of 0.1 is used on the parameter updates to avoid filter inbreeding and keep the ensemble spread larger so that the model error covariance matrix is a good approximation for model uncertainty.

In previous studies parameters were updated indirectly (Naz et al., 2019; Han et al., 2014; Baatz et al., 2017). We tested directly updating saturated hydraulic conductivity, porosity, hydraulic conductivity

exponent, and saturated soil matric potential but this resulted in more unstable estimates than indirectly updating soil hydraulic parameters. The pedotransfer function which is used for the indirect updating results in reasonably correlated soil hydraulic parameters. In testing a direct approach to updating saturated hydraulic conductivity, porosity, hydraulic conductivity exponent B, and saturated soil matric potential we found that updating the parameters indirectly to provide more stable simulations. The pedotransfer function keeps the soil hydraulic parameters reasonably correlated to each other. In this study, the parameters are chosen to optimize the SWC estimation and not ET estimation to study the effects of SWC improvements on ET. To more directly improve the ET estimation, parameters that are critical to the ET process should be added, e.g. vegetation hydraulic parameters that are related to the transfer of water from the root to leaf or parameters related to stomatal conductance.

## 2.4 Model setup

### 2.4.1 Domain setup

Since we only use local field measurements, we represent each study site as a single grid cell in CLM5. This approach is also consistent from the viewpoint of larger regional scale models, where each of these sites would only be part of a grid cell. The CLM5 grid cells are vertically divided into 25 layers from the surface down to 50 m depth of which the first 20 layers (until 8.6 m depth) may be hydrologically and biogeochemically active depending on the variable soil depth for each site (Lawrence et al., 2018). For the more than 70 different surface parameters of CLM5, we used the default values generated by the tools provided with CLM5 (e.g. soil depth to bedrock, sand, clay, and organic matter fractions, PFTs). These default values are generated from remapping various global files (Lawrence et al., 2019). Only the PFT were manually assigned for each site. For the ensemble creation, the fractions of sand, clay, and organic matter are modified for each ensemble member. The perturbations are normally distributed with mean zero and a standard deviation of 10%.

### 2.4.2 Atmospheric forcings

Meteorological observations were also available at the selected study sites and were used to force CLM5. The existing gaps in the observation time series were gap-filled with data from the COSMO-REA6 reanalysis data product (Bollmeyer et al., 2015). For the ensemble generation precipitation (PR), shortwave radiation (SW), longwave radiation (LW), and air temperature (TA) were perturbed taking into account cross-correlations between variables according Reichle et al. (2007). The perturbations are: multiplicative PR ~ logN(1, 0.5), multiplicative SW ~ logN(1, 0.3), additive LW ~ N(0, 20) (W/m2), and additive TA ~ N(0, 1) (°K). The following cross-correlation coefficients between variables were used: PR–SW -0.8, PR-LW 0.5, PR-TA 0, SW-LW -0.5, SW-TA 0.4 and LW-TA 0.4.

### 2.4.3 Data assimilation experimental setups

Three different simulation scenarios were considered: 1) Open loop (OL) simulations without data assimilation; 2) Data assimilation with updating of soil water content (DAS); and 3) Data assimilation with soil water content updating and parameter updating (DASP). For all scenarios data assimilation is performed at a daily frequency and with daily averages from the observations. The observation error is assumed to be constant and set to a RMS of 2%.

### 2.5 Statistical metrics

For the comparison of simulation results with observations, we use four statistical metrics: the squared correlation coefficient ($R^2$), the mean bias error (MBE), the root-mean-square error (RMSE), and the unbiased root-mean-square-error (ubRMSE):

$$R^2 = 1 - \frac{\sum_{t=1}^{Nt}(o^t - m^t)^2}{\sum_{t=1}^{N}\left(o^t - \overline{o^t}\right)^2} \quad (7)$$

$$MBE = \frac{\sum_{t=1}^{Nt}(m^t - o^t)}{Nt} \quad (8)$$

$$RMSE = \sqrt{\frac{\sum_{t=1}^{Nt}(m^t - o^t)^2}{Nt}} \quad (9)$$

$$ubRMSE = \sqrt{\frac{\sum_{t=1}^{Nt}\left[\left(m^t - \overline{m^t}\right) - \left(o^t - \overline{o^t}\right)\right]^2}{Nt}} \quad (10)$$

where o stands for observations, m represents the ensemble average of the simulated values, t is the time step, Nt the total number of time steps and overbar represents the average over all time steps.

### 3. Results

#### 3.1 Soil water content and related parameters

Figures 2 and 3 show the results of the soil water content simulations at 20 cm depth of the OL, DAS and DASP simulations compared to the soil water content observed at the nine sites. Figure 2 compares the OL and DAS results and Fig. 3 compares the OL and DASP results. The corresponding scatter diagrams for the depths 5, 20, and 50 cm can be found in the appendix (Figures A1 - A7). Overall, the results show expected improvements by data assimilation of observed soil water content. For the OL simulations, Fig. 2 shows particularly large RMSE values for CZ-BK, DE-Obe, FI-Sod and NL-Loo. Fig. 2 also illustrates the improved performance achieved by DASP, with a RMSE reduction from 29.3 cm³/cm³ to 6.25

cm³/cm³ and a MBE reduction from 28.06 cm³/cm³ to -2.94 cm³/cm³ for FI-Sod. Parameter updating, as shown in Fig. 3, further improves the simulation results, but the improvement from DAS to DASP is significantly less than from OL to DAS.

The results of the three COSMOS-Europe sites are shown in Figure 4, in which the observed SWC values are compared with the weighted SWC mean of the model layers corresponding to the measurement depth of CRNS. This comparison again shows the large improvement from OL to DAS, and a smaller improvement or even a small deterioration from DAS to DASP.

Figure 5 shows the depth profile for the root fraction and the SWC average of the OL and DASP simulations for the first 1.2 meters (10 layers) for each site. The SWC is updated for all layers, including the layers with the largest root fraction, but depending on the site the magnitude of the update varies with depth. For most sites the data assimilation shifts the SWC values while keeping the profile similar to the OL results. FI-Hyy and FI-Sod are the exception and show a decrease of SWC in the first 25 to 50 cm and an increase of SWC in the deeper layers for DASP.

Figure 6 shows time series of the estimated saturated soil hydraulic conductivity for each of the sites and the three observation layer depths. The DASP scenario results in parameter changes when the first observations are available but converge over the time of the simulation to a new value. The corresponding time series for the other soil hydraulic parameters can be found in the appendix (Figures A8, A9, and A10).

Figures 7, 8, and 9 show the initial (prior) and the updated (posterior) vertical profiles for the sand-, clay-, and organic matter fractions for the upper 1.2 meters (10 soil layers). The updated parameters often keep the profile distribution but have reduced or increased values throughout the layers compared to the prior.

## 3.2 Evapotranspiration

The impact of the data assimilation on the ET flux is shown in Figures 10 and 11. Notably, the difference between the OL and the DASP results is smaller for ET than for SWC. While the data assimilation improves the model results for SWC for all sites, both improvement and deterioration occur for modeled ET. Figure 12 shows the comparison of the improvements by data assimilation for SWC and the positive and negative effect on ET estimation. The average RMSE reduction for the DASP SWC prediction is between 56% and 64% compared to OL. Comparing the OL and DASP results for ET shows an average reduction of the MBE of 0.06 mm/day, but an increase in RMSE for the DASP ET predictions of 4 % on average, with 8 of the 13 sites showing a relative change in ET of only +/- 1 %. Two outliers (FI-Sod and NL-Loo sites) reduce the average model improvement. These sites show both a large overestimation in SWC in the OL (see Fig. 2) and a large underestimation of ET in the DASP simulation (see Fig. 11). This

could be caused by the mismatch of simulated and actual LAI for these sites. To investigate this, we repeated the simulations using CLM5 with satellite-derived phenology (CLM5-SP) the results are shown in Fig. 13. Because the focus of this study is on CLM5-BGC, these CLM5-SP simulations use the default datasets from CLM5 since in-situ LAI measurements for these sites were not available The CLM5-SP OL and DASP simulations do not use any information from the CLM5-BGC simulations which implies that for the CLM-SP DASP simulations parameters are estimated independently from the CLM5-BGC simulations. For CLM5-SP we observe an average improvement in the RMSE of SWC between 57.6 % and 64.3 % and an average reduction of 5.8 % for the ET estimation. These CLM5-SP simulations use the default datasets from CLM5 and without site specific calibration of the timing or magnitude of the seasonal phenology of LAI. Therefore, even for the CLM5-SP simulations, there is a mismatch between simulated and actual LAI. However, also for this case there are sites with large improvement in SWC estimation that show deterioration for ET estimation.

Another possible explanation for the improvement in SWC estimation but no improvement of ET estimation is the underestimation of root water uptake from deeper soil layers for forest sites, as also suggested by Shrestha et al. (2018). Fig. 12 shows that the quality of the model results is not dependent on the forest type, i.e. the evergreen needle leaf forests (ENF) sites show both strong and average relative changes in SWC RMSE and ET RMSE. This suggests that the strong deviations in the model results of the FI-Sod and NL-Loo sites are due to other local conditions, e.g. soil properties.

The three CRNS sites show an average relative change of ET RMSE of -2.6%, -0.2%, and -0.9% for DE-HoH, DE-Wue, and DK-Glu, respectively. Therefore, although the CNRS measurements are more consistent with the large measurement area of the flux towers, no significant improvement in ET for these three sites can be achieved with the current implementation of the CNRS-SWC assimilation. We anticipate that the implementation of a more accurate observation operator would improve the modeled SWC. The current observation operator does not use vertical weighting to take the decreasing CRNS sensitivity with depth into account.

### 3.3. Evaluation of other land-atmosphere exchange fluxes

Comparing measured and modeled sensible heat fluxes (SH) are compared to measurements (Figures 14 and 15), similar $R^2$ values are obtained for the OL and the DASP approach. The $R^2$ values range from 0.23 to 0.51 with an average of 0.36. This is similar to the ET results, where the $R^2$ of measured and modeled (OL and DASP) ET range from 0.01 to 0.58 with an average of 0.37. Comparing Fig. 14 and Fig. 15 shows the impact of data assimilation of SWC on SH to be small. On average DASP improves

the MBE by 4.66 W/m² compared to OL. However, for five of the eight sites the improvement of the MBE is smaller than 1 W/m². But, compared to the ET results data assimilation of SWC reduces the MBE of SH for all sites.

The impact of updating SWC with data assimilation on modeled Nee, GPP, and LAI is shown in Fig. 16. The NEE is negative (land acts as carbon sink) for eight, seven, and six of the field sites for OL, DAS, and DASP respectively. For DASP the GPP and LAI show an increase for two of the sites, a decrease for three of the sites and remain similar for eight of the sites. Fig. 17 shows how average SWC in 5 cm and 50 cm depth, ET, NEE, GPP and SH (average over all sites and all years) are affected by data assimilation. Although DASP adjusts SWC at 5cm towards the observations, the correction for SWC at 50cm depth is smaller because not all sites provide data at this depth. However, for all sites the data assimilation provides some improvement for SWC estimation even in layers below the observation depth. In spite of improved SWC characterization, ET deviates slightly more from the observations after DASP, while sensible heat flux is very slightly closer to the observations. GPP is lower after DASP and NEE less negative. While the overall change for some of these variables is small, different variations throughout the year can be observed. This averaging hides the variations between sites and annual variability but highlights the overall model behavior. Notably, the data assimilation improves SWC estimation at 5cm throughout the year while at 50 cm depth the improvement can mainly be observed in late summer and autumn. Similarly, for SH a model structural bias is apparent with large negative simulated SH values in late autumn, winter, and early spring while the observations show only a few days with negative average values over all sites and all years.

Figure 18 shows the LAI for each site averaged over all the simulated years and the difference between the prescribed LAI used in CLM5-SP and the simulated LAI by CLM5-BGC. Sites with the same PFT show clear differences in the yearly LAI cycle.

# 4. Discussion

## 4.1 Soil water content improvements

Our results confirm that assimilation of high quality in-situ SWC data improves the prediction of SWC by CLM5, as it has been demonstrated in several other studies (Hung et al., 2022; Mahmood et al., 2019; Naz et al., 2019; Liu and Mishra, 2017; Han et al., 2015 ). In our study, we were able to show that this also applies to forest sites with different climates, tree species, and soil properties.

Additionally, CRNS observations represent SWC for a larger area in better correspondence to the EC tower footprint. So far, only few studies have used CRNS information in a data assimilation framework (Rosolem et al., 2014; Han et al., 2015; Baatz et al., 2017, Patil et al., 2021). In line with our study, these studies show the high potential of CRNS for improved soil moisture prediction with land surface models, both in terms of SWC prediction as well as in improving soil hydraulic parameters. Currently, CRNS stations are operated with increasing numbers worldwide (Andreasen et al., 2017), in hydrological observatories (e.g. Bogena et al., 2018; Lui et al., 2018), as national networks (Zreda et al., 2012; Evans et al., 2016) or even increasing at continental scales (e.g. Hawdon et al., 2014; Bogena et al., 2022), which opens up new opportunities for assimilation of CRNS data in land surface models at various scales.

In our data assimilation approach, we assumed that the CRNS signal shows a constant sensitivity to SWC down to the penetration depth of the CRNS. However, Schrön et al. (2017) have shown that the integrated neutron signal over a vertical soil column exhibits a strong decrease in sensitivity with depth and suggested that this physical behavior of neutrons should be taken into account in model applications. For example, Shuttleworth et al. (2013) developed a simple, physically-based analytical model to translate model-predicted soil moisture profiles into aboveground fast neutron counts within a data assimilation framework. A simpler method was proposed by Schrön et al. (2017) by using vertical weighting functions that depend on SWC, atmospheric pressure, horizontal distance and vegetation height. Therefore, in a follow-up study, we will test whether observation operators that account for the vertical weights of the different model soil layers according to the decreasing sensitivity of CRNS with depth will improve our SWC prediction results.

## 4.2 Evapotranspiration estimation without improvements from SWC DA

Several studies have demonstrated the potential of improved ET prediction using data assimilation of SWC measurements (Liu and Mishra, 2017; Girotto et al., 2017; Lidard et al., 2011). These studies focused on regional or global scale and show heterogeneous spatial patterns of improvement to ET estimation. Baatz et al. (2017) showed that assimilation of CRNS observations altered the ET estimation in CLM4.5 in parts of their study area by up to 80 mm per year compared to the OL approach.

However, in our study with the land surface model CLM5 that data assimilation of SWC does not improve the ET prediction for European forest sites. We also found that the impact on ET from assimilating CRNS observations is similarly limited as assimilation of other in-situ SWC data Since our study sites cover a variety of climates and soil types, we assume that this result also applies to other forest sites worldwide with similar tree species.

The lack of improvement in ET prediction in the case of data assimilation of in-situ soil moisture information is consistent with findings from other studies. Girotto et al. (2017) found a decrease in ET accuracy after assimilating GRACE data over India and attributed the results to the representation of irrigation in the model. Similarly, Peters-Lidard et al. (2011) showed mixed results after assimilating multiple satellite soil water content products over North America with spatial variation of improvements and deterioration of ET estimation. Overall, for 9 of the 13 forested study sites our OL simulations show positive mean bias error indicating that CLM5 underestimates the ET compared to the FLUXNET observations. These underestimations agree with the results shown in the study by Cheng et al. (2021) showing that CLM5 underestimates ET observations. Additionally, Nearing et al. (2018) investigated the contribution of model structural errors and model inputs for four different LSM and concluded that SWC uncertainty was dominated by soil parameter uncertainty while ET uncertainty was dominated by forcing uncertainty. Without a similar in-depth benchmark study for CLM5, but from our results and the results of the previously mentioned CLM5 studies a similar conclusion can be drawn for CLM5.

A different aspect is that we assume that the EC data are correct to validate our simulation results. However, the EC-data might be affected by energy balance closure issues (Foken, 2008; Hendricks Franssen et al., 2010).

## 4.4 Methods to improve ET estimation

There are various approaches to improve modeled ET estimates. For example, Zhang et al. (2020) identified and optimized four hydraulic and three vegetation parameters in CLM4.0 that improved ET estimation by 7.3% for the optimization period and 5.3% for the validation period for China. Similarly, Post et al., (2017) calibrated eight parameters to improve NEE estimation in CLM 4.5 and a similar approach to optimize vegetation parameters in CLM 5.0 for ET estimation could improve simulation results. Tang et al. (2015) implemented a root hydraulic redistribution model in CLM4.5 to improve ET estimation but found that their method was only able to improve ET predictions north of 20°N. They identified the representation of deep roots, soil hydraulic parameterization for certain soils, meteorological forcings, and the parameterization of the water table dynamics and drainage as the main limitations to improve ET by their method.

Denager et al. (2022) used SWC measurements for an agricultural site in Denmark for parameter calibration of soil texture, LAI, stomata conductance and the root distribution in CLM5 and obtained improved energy partitioning of ET and SH. However, they also found it difficult to calibrate the

parameters to get an improvement in SH estimation throughout the year and suggested that the difference in energy balance closure between LSMs and EC flux observations contributes to the bias.

Fox et al. (2022) concluded that errors in LAI estimations in LSMs lead to substantial flaws in the representation of carbon, water, and energy fluxes. Furthermore, they conclude that data assimilation to remove bias in LAI improves LSMs results significantly and is advisable until the prognostically modeled LAI improves substantially. For example, Zhang et al. (2016) assimilated remotely sensed LAI data into the Biome-BGC model at two sites and improved both ET and NEE estimates, evaluated with EC tower measurements. Rahman et al. (2022) showed that the joint LAI and topsoil SWC assimilation from satellite products improved the ET estimation for the Contiguous United States compared with independent validation datasets. While data assimilation of topsoil SWC alone only improved the SWC estimation.

As mentioned, LAI is identified as a key variable to improve ET estimation and representation of land carbon processes. Therefore, in future work we will investigate the effects of data assimilation of LAI and joint state-vegetation parameter estimation on the simulation of carbon, water, and energy fluxes with CLM5.

## 5. Conclusions

This paper analyzed the impact of the assimilation of in situ soil water content (SWC) data on SWC characterization, evapotranspiration (ET), sensible heat flux (SH), gross primary production (GPP) and net ecosystem exchange (NEE), for 13 forested sites in Europe. Assimilation of SWC, from both point scale and plot scale observations, with the Ensemble Kalman Filter, using the Community Land Model version 5 coupled to the Parallel Data Assimilation Framework (CLM5-PDAF) improves SWC prediction (RMSE reductions between 56% and 64% compared to the open loop run, and depending on measurement depth). However, assimilation of in situ SWC does not improve the ET prediction for the investigated European forest sites. For most of the sites, data assimilation showed almost no effect on ET fluxes (RMSE changes between +/- 1%) and some sites showed strong negative effects of SWC assimilation on ET predictions ( -20% to -30% change in RMSE). The assimilation of in situ SWC from Cosmic Ray Neutron Sensors (CRNS), which determine SWC over a larger horizontal footprint more in correspondence with the eddy covariance footprint, for three of the 13 sites, also does not improve ET characterization. These results suggest that improving the SWC estimation of state-of-the-art LSM such as CLM5 is not sufficient to improve ET estimation for forest sites. To improve ET estimation it is also necessary to consider the representation of LAI in magnitude and timing, as well as uncertainties in water uptake by roots and vegetation parameters In the future, to improve modeled ET using data assimilation we will further examine the potential of assimilating different state variables, like for example leaf area index and updating related vegetation parameters. In addition, we will apply a measurement operator in the data assimilation framework that considers the vertical sensitivity of the CRNS signal.

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

## Tables

**Table 1**: Overview of the study sites. Classification uses the International Geosphere-Biosphere Program Code (IGBP) as is used for FLUXNET: MF for mixed forests, ENF for evergreen needle leaf forests, DBF for deciduous broad leaf forests, EBF for evergreen broad leaf forests, WSA for woody savannah. LON is longitude and LAT latitude.

| Site name | Country | Abbreviation | Code | LON | LAT | Elevation [m.a.s.l.] | Data source | Mean annual temperature [°C] | Mean annual precipitation [mm] | Typical tree species |
|-----------|---------|--------------|------|-----|-----|----------------------|-------------|------------------------------|---------------------------------|----------------------|
| Brasschaat | Belgium | BE-Bra | MF | 4.51 | 51.3 | 16 | FLUXNET | 9.8 | 750 | Scots pine |
| Bílý Kříž forest | Czech Republic | CZ-BK | ENF | 18.53 | 49.5 | 875 | FLUXNET | 7 | 1316 | Norway spruce |
| Hainich | Germany | DE-Hai | DBF | 10.45 | 51.07 | 430 | FLUXNET | 8.3 | 720 | Mixed Beech |
| Hohes Holz | Germany | DE-HoH | DBF | 11.21 | 52.08 | 217 | COSMOS Europe | 10 | 820 | Mixed beech |
| Oberbärenburg | Germany | DE-Obe | ENF | 13.72 | 50.78 | 734 | FLUXNET | 5.5 | 996 | Norway spruce |
| Wüstebach | Germany | DE-Wue | ENF | 6.33 | 50.5 | 605 | COSMOS Europe | 7 | 1180 | Spruce |
| Gludsted | Denmark | DK-Glu | ENF | 9.33 | 56.07 | 86 | COSMOS Europe | 8.2 | 1080 | Spruce |
| Conde | Spain | ES-Cnd | WSA | -3.22 | 37.91 | 370 | FLUXNET | 15.8 | 474 | Olive grove |
| Hyytiälä | Finland | FI-Hyy | ENF | 24.29 | 61.84 | 181 | LTER Europe | 3.8 | 709 | Boreal Scots pine |
| Sodankylä | Finland | FI-Sod | ENF | 26.63 | 67.36 | 180 | FLUXNET | -1 | 500 | Boreal Scots pine |
| Puéchabon | France | FR-Pue | EBF | 3.59 | 43.69 | 270 | FLUXNET | 13.5 | 883 | Evergreen oak |
| Lavarone | Italy | IT-Lav | ENF | 11.28 | 45.95 | 1353 | FLUXNET | 7.8 | 1291 | Coniferous forest |
| Loobos | Netherlands | NL-Loo | ENF | 5.74 | 52.16 | 25 | FLUXNET | 9.8 | 786 | Scots pine |

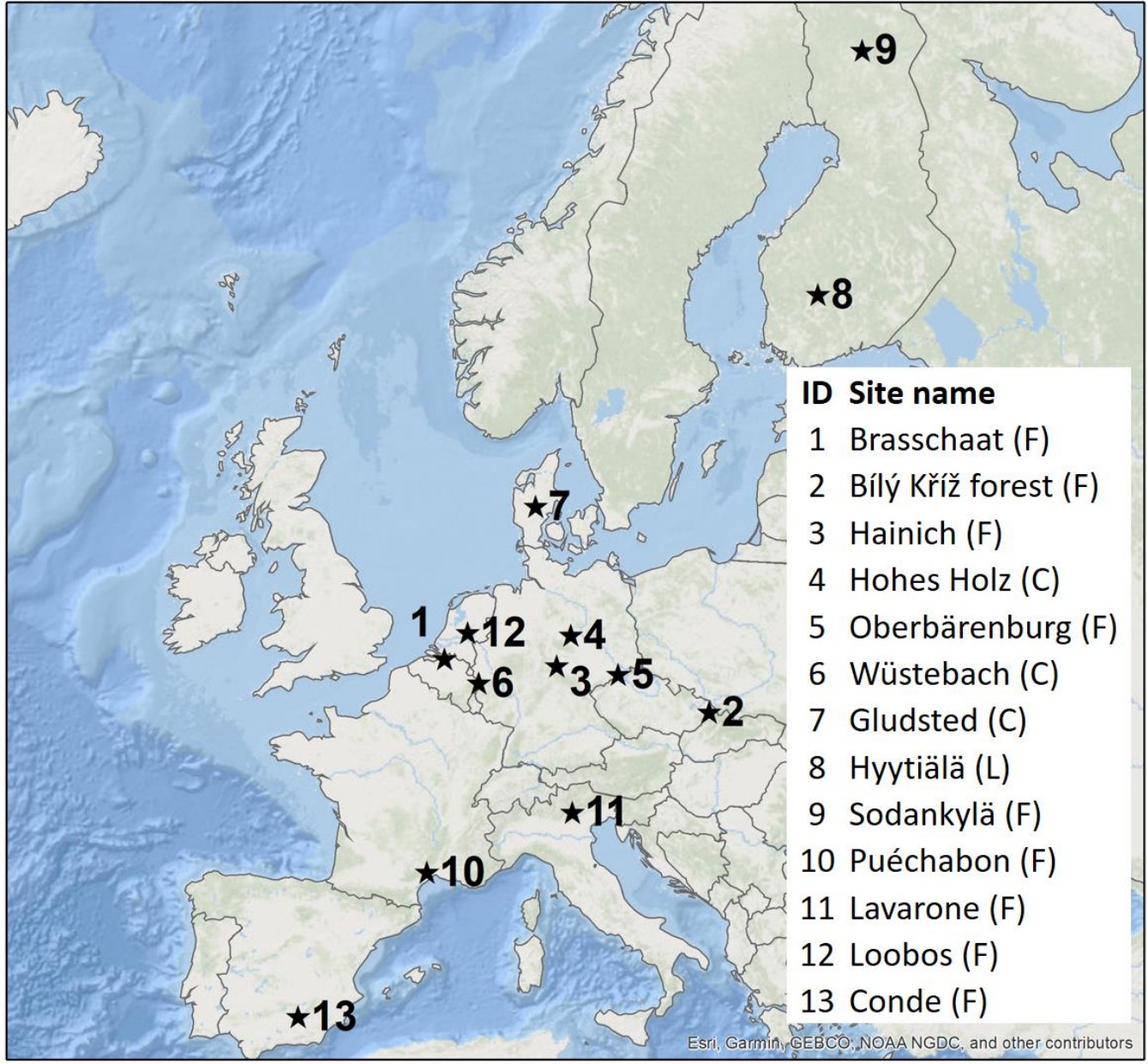

**Figure 1:** Map showing the location of the selected study sites of the FLUXNET (F), eLTER (L) and COSMOS-Europe (C) networks.

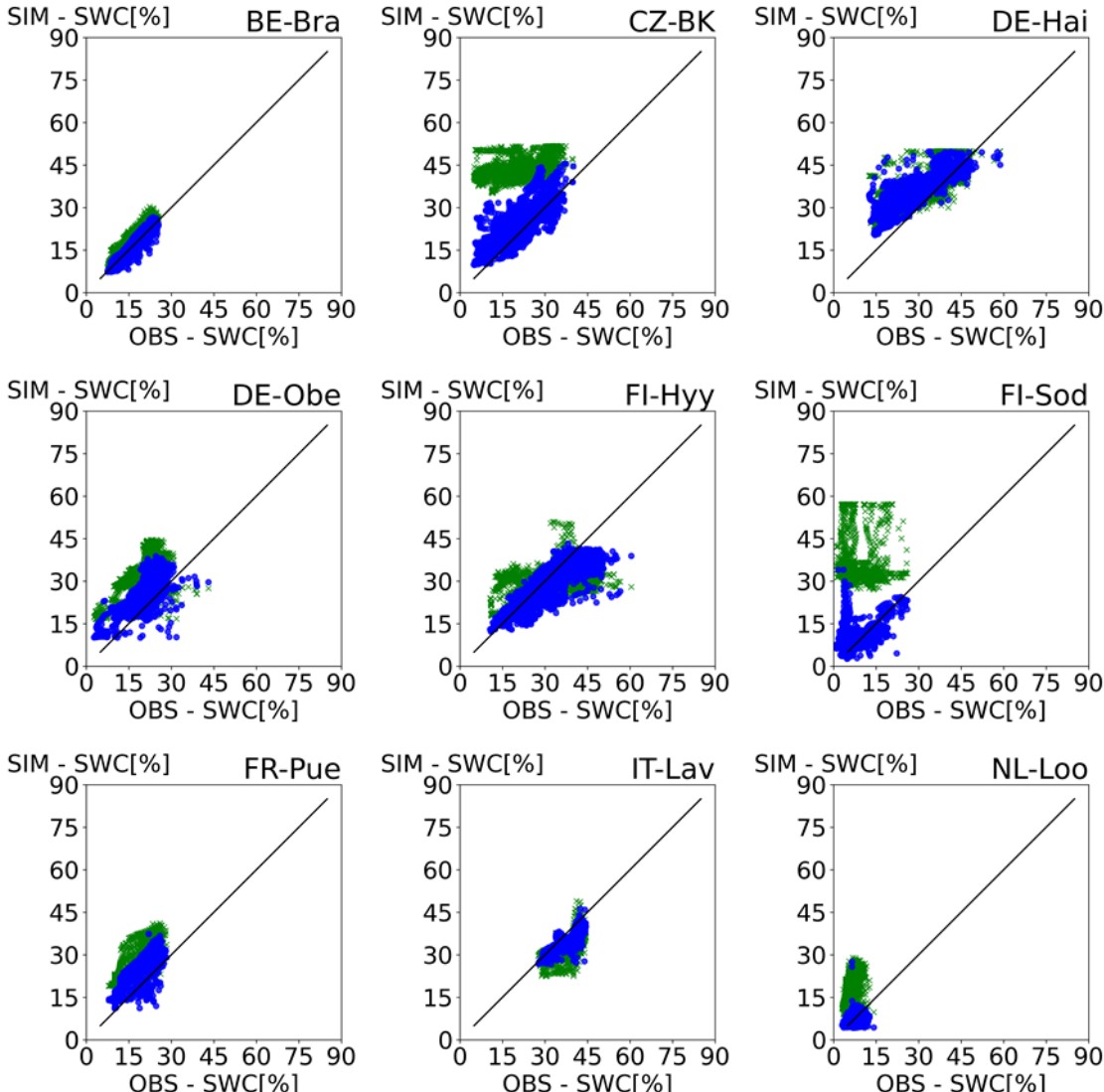

**Figure 2:** Scatter plots of observed soil water content at 20 cm depth at nine study sites versus OL and DAS simulated soil water content. The points represent daily averages for the days observation data are available. Green points are OL and blue points are DAS results.

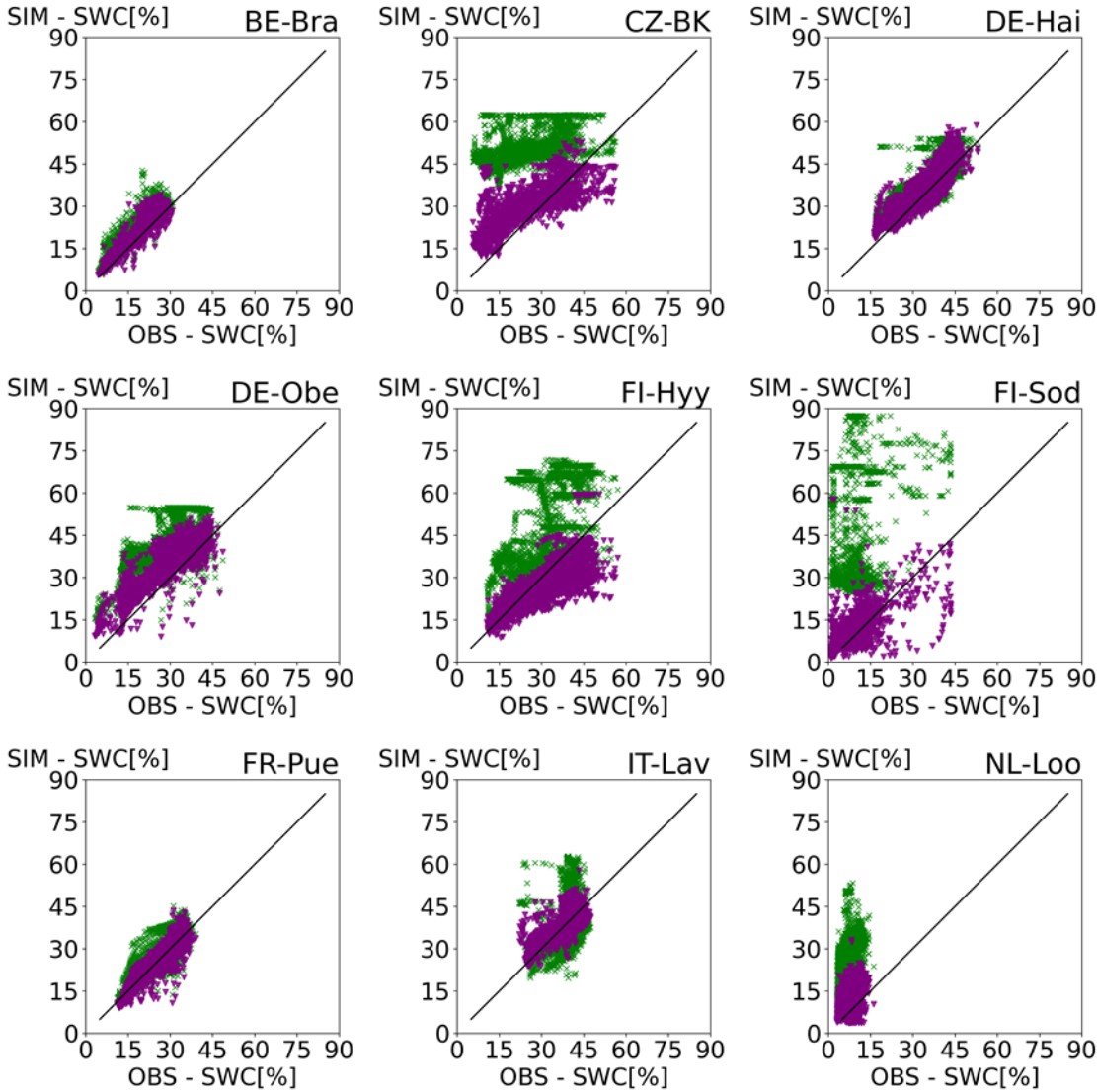

**Figure 3:** Scatter plots of observed soil water content at 20 cm depth at nine study sites versus OL and DASP simulated soil water content results at 20 cm depth. The points represent daily averages for the days observation data are available. Green points are OL and purple points are DASP results.

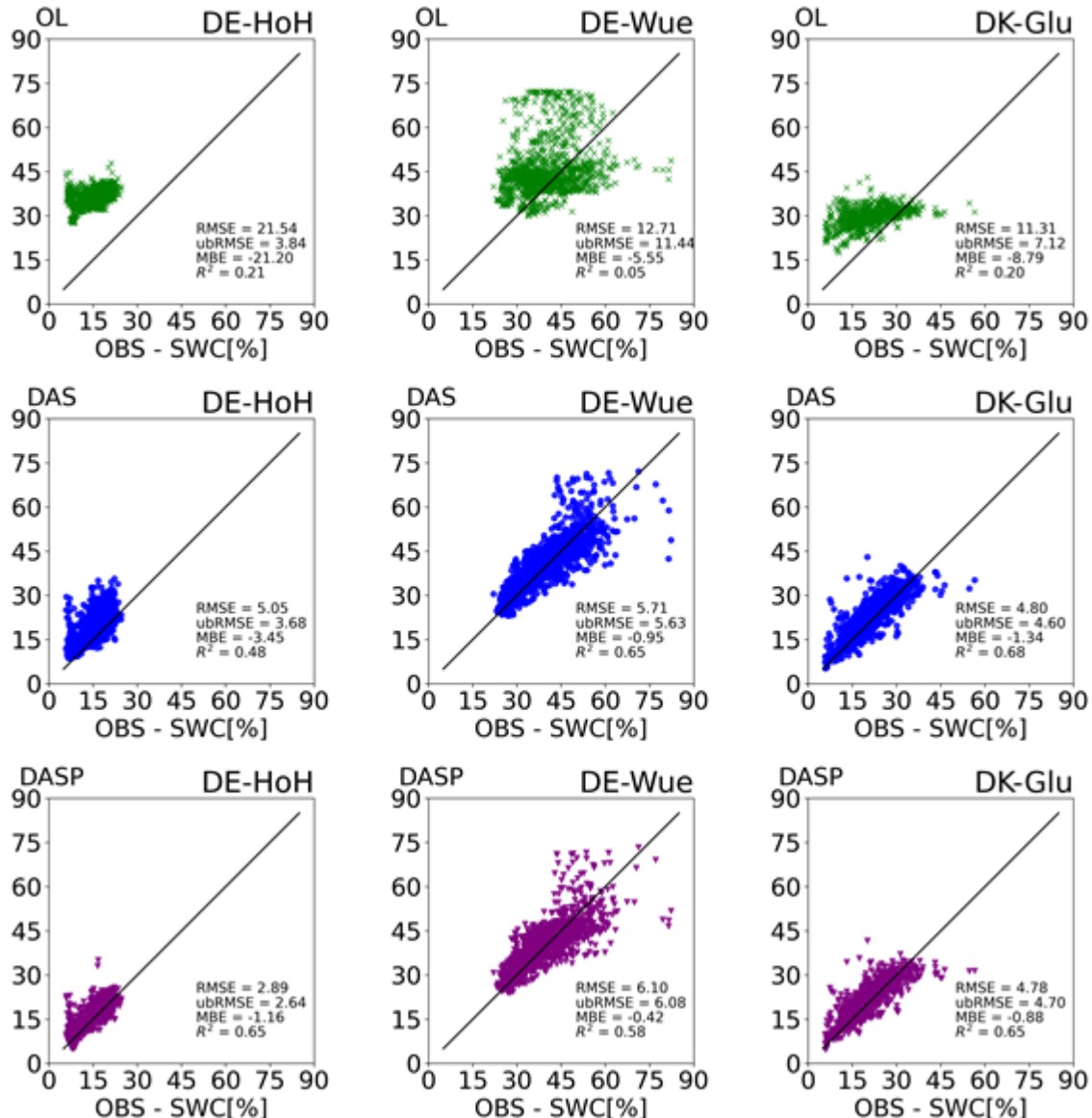

**Figure 4**: Scatter plots of observed soil water content at three CRNS study sites (DE-HoH left column, DE-Wue middle column, DK-Glu right column) versus simulation results (OL results in the top row, DAS results in the middle row, and DASP results in the bottom row). The points represent daily averages for the days on which observation data are available.

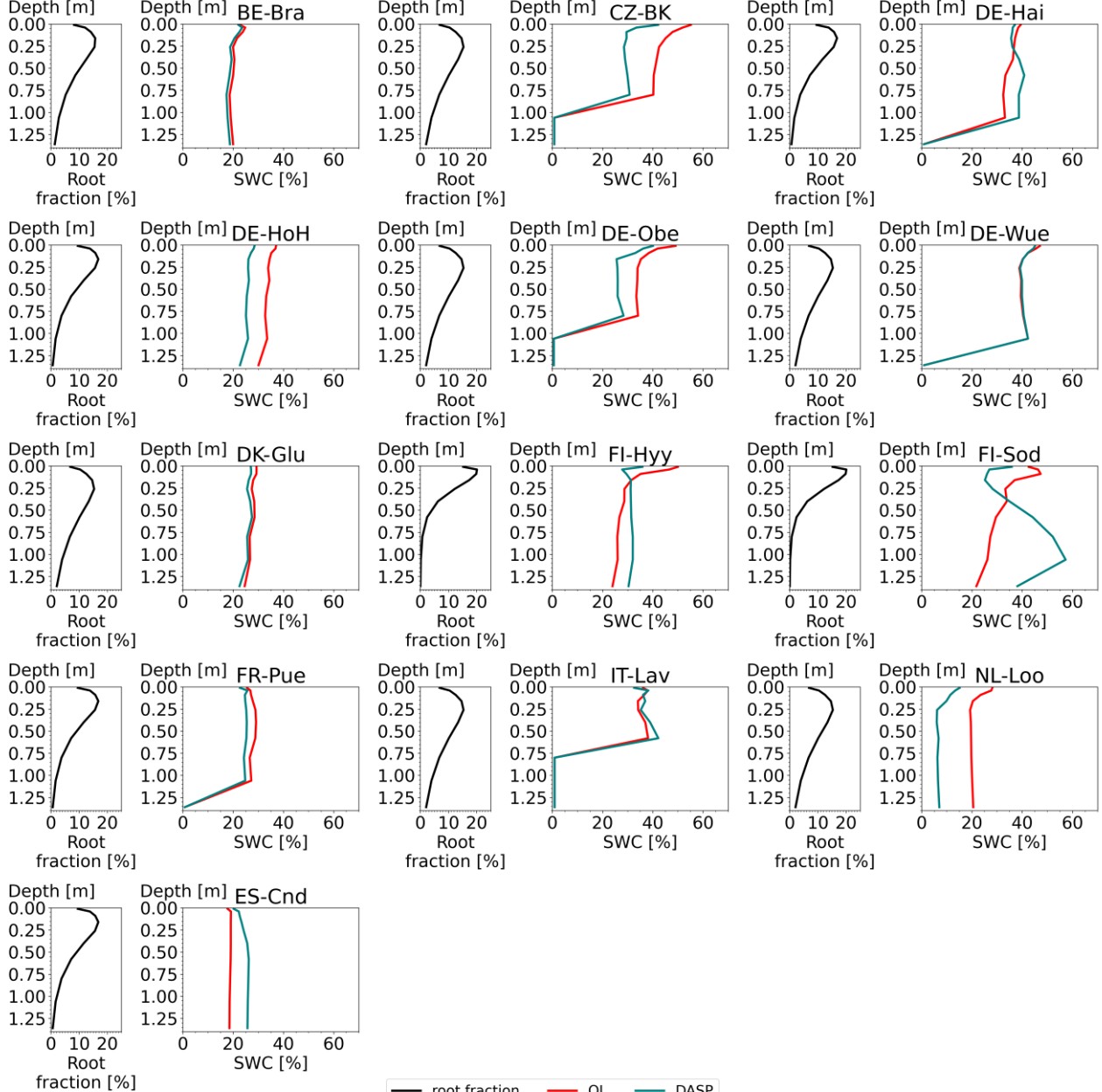

**Figure 5**: Profile plots for the first 10 layers, showing the root fraction and the time-averaged SWC per depth for each site. In the SWC profiles, the red and green lines represent the SWC from the open-loop simulations (OL) and DASP simulations, respectively.

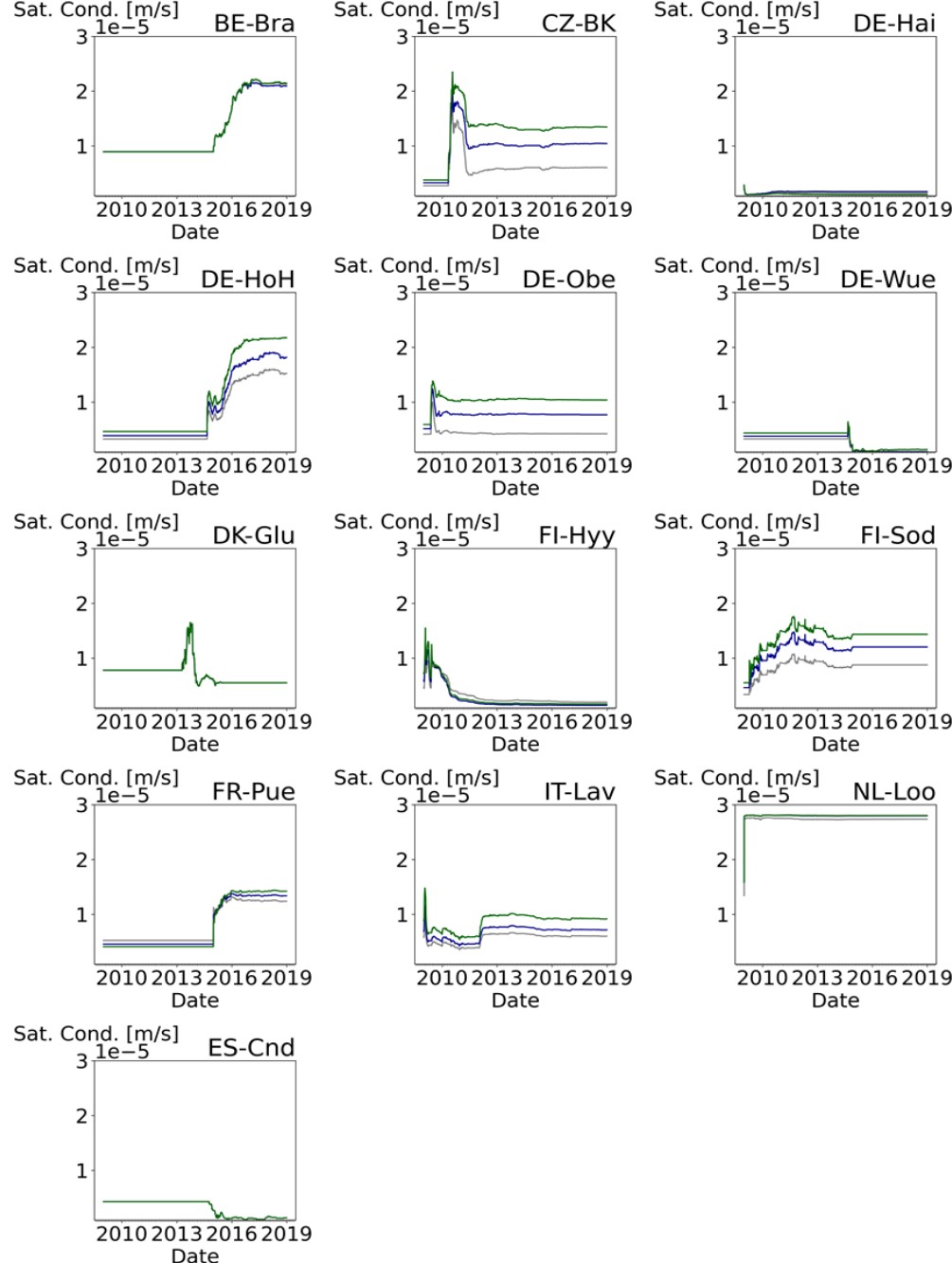

**Figure 6:** Time series of the saturated soil hydraulic conductivity for each site in the DASP simulation. The grey line is the value at 5 cm depth, the blue line at 20 cm depth, and the green line at 50 cm depth.

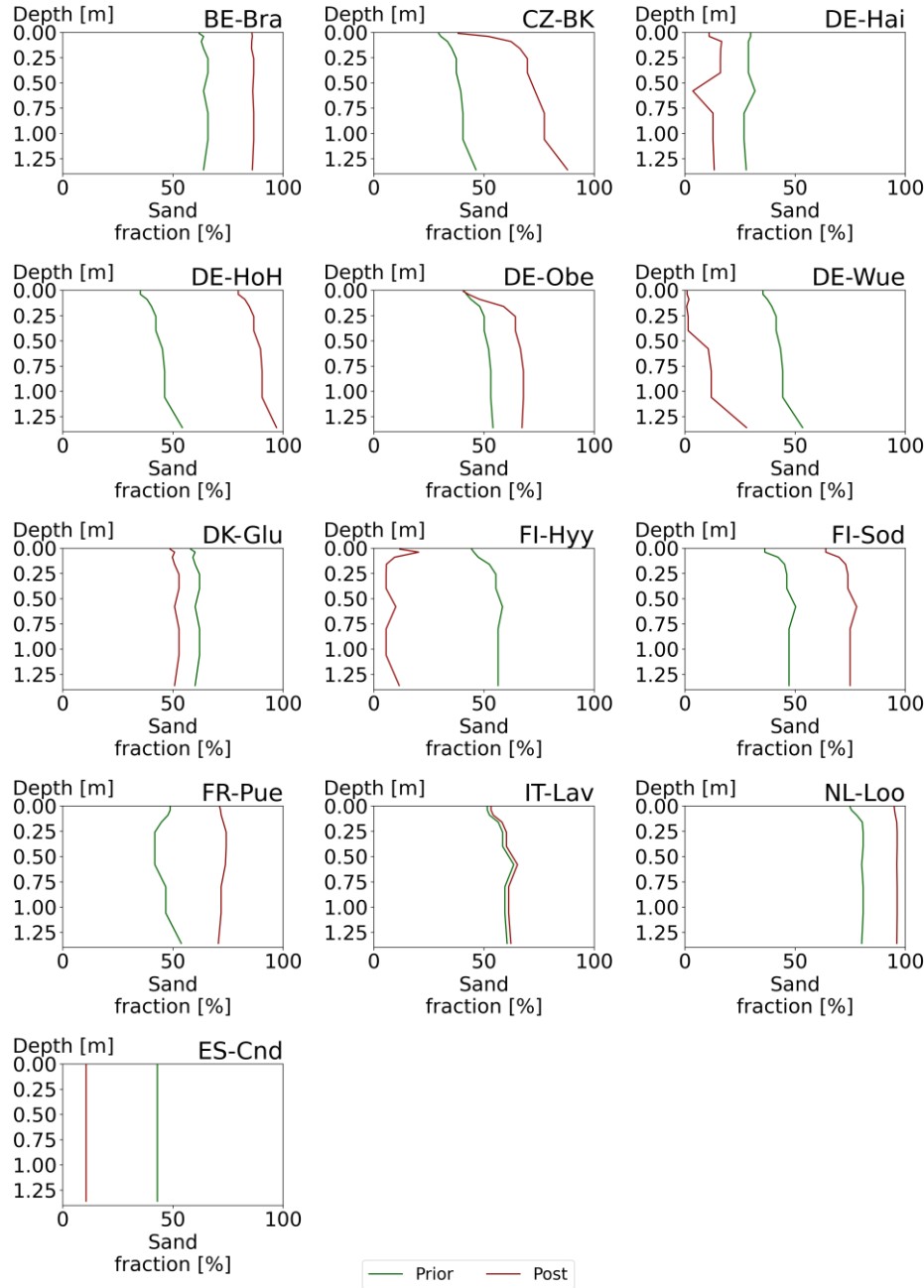

**Figure 7**: Profile plot showing the sand fractions for the first 10 layers of all 13 sites.

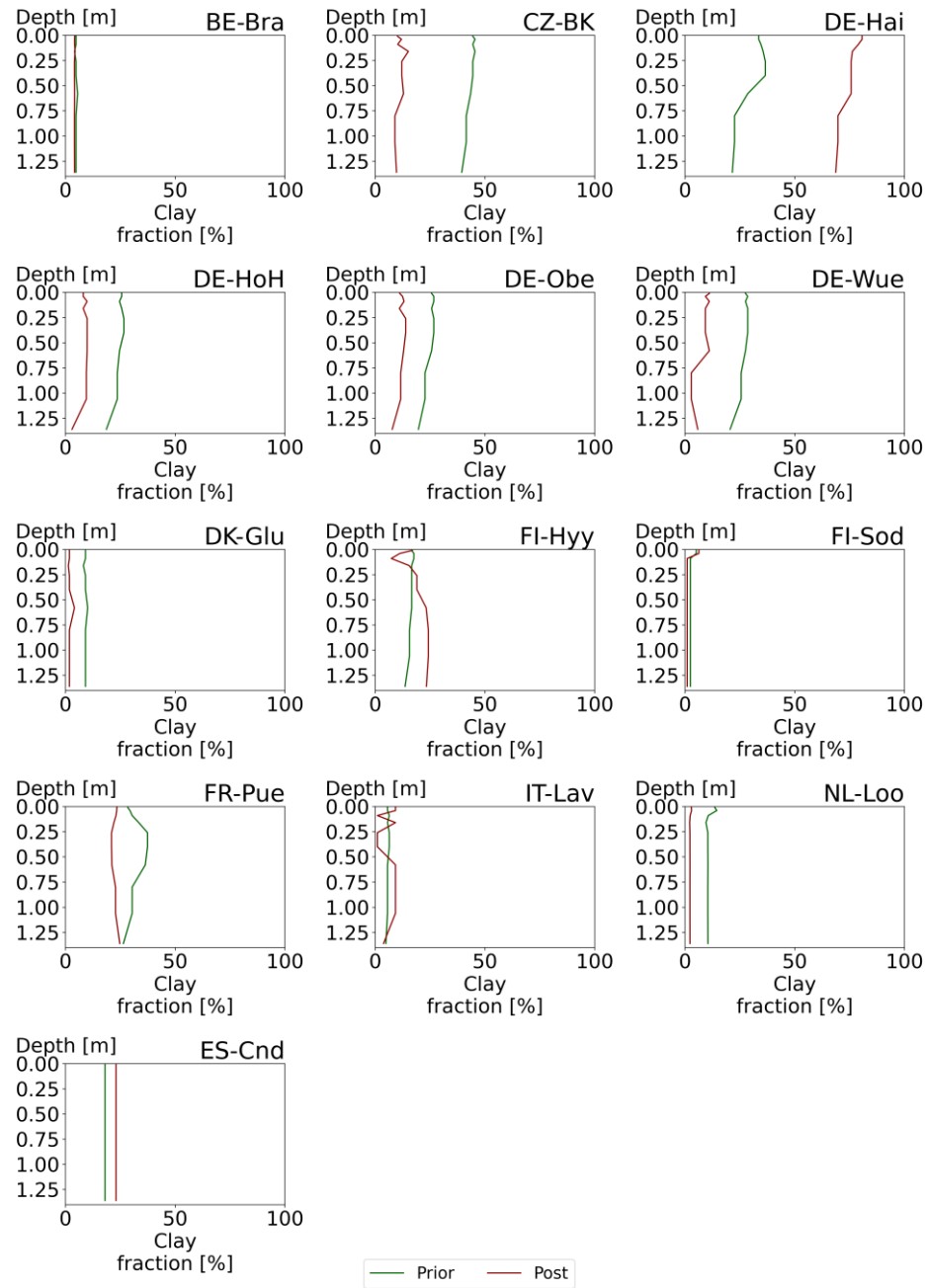

**Figure 8:** Profile plot showing the clay fractions for the first 10 layers of all 13 sites.

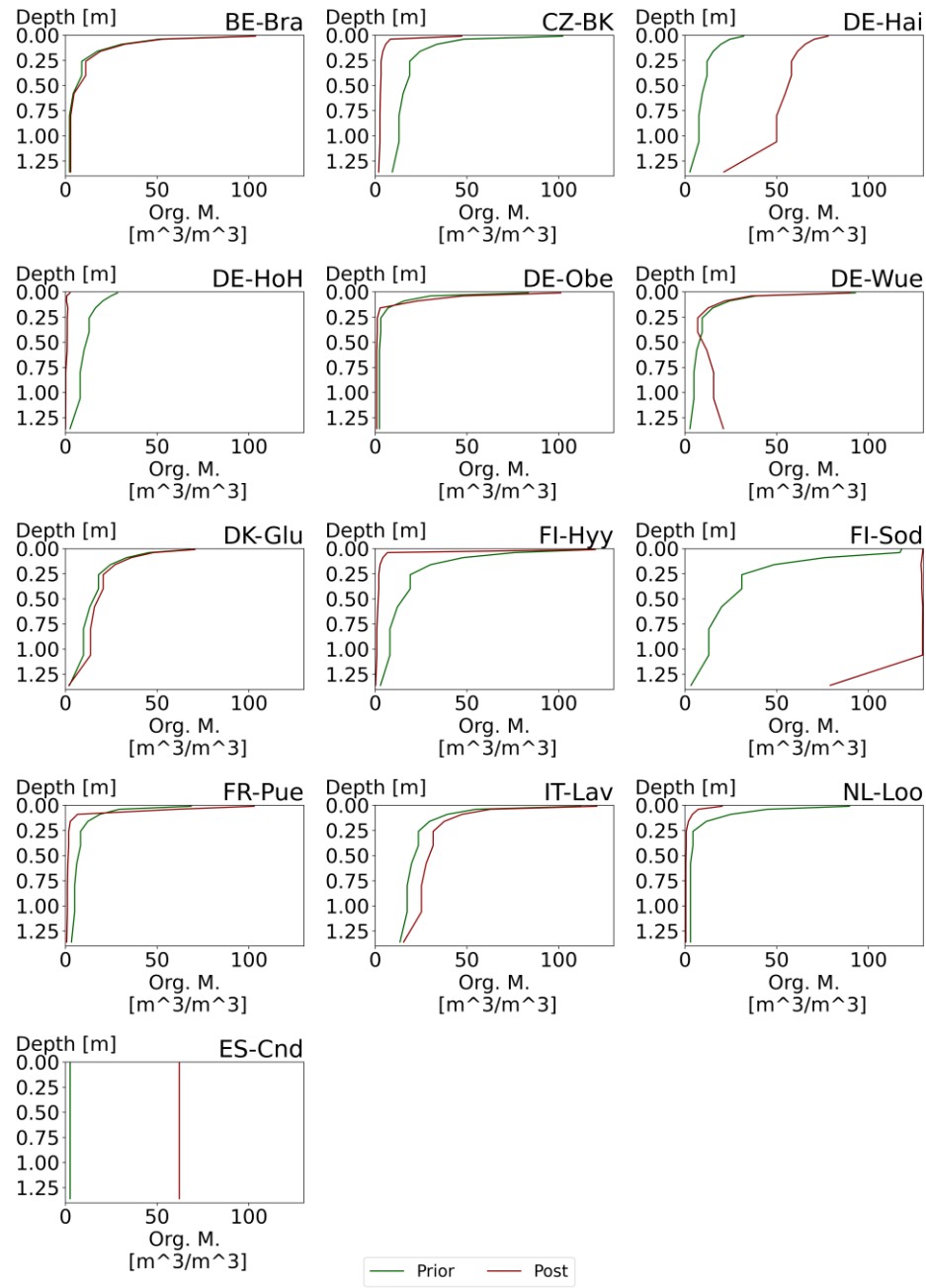

**Figure 9**: Profile plot showing the organic matter fractions for the first 10 layers of all 13 sites.

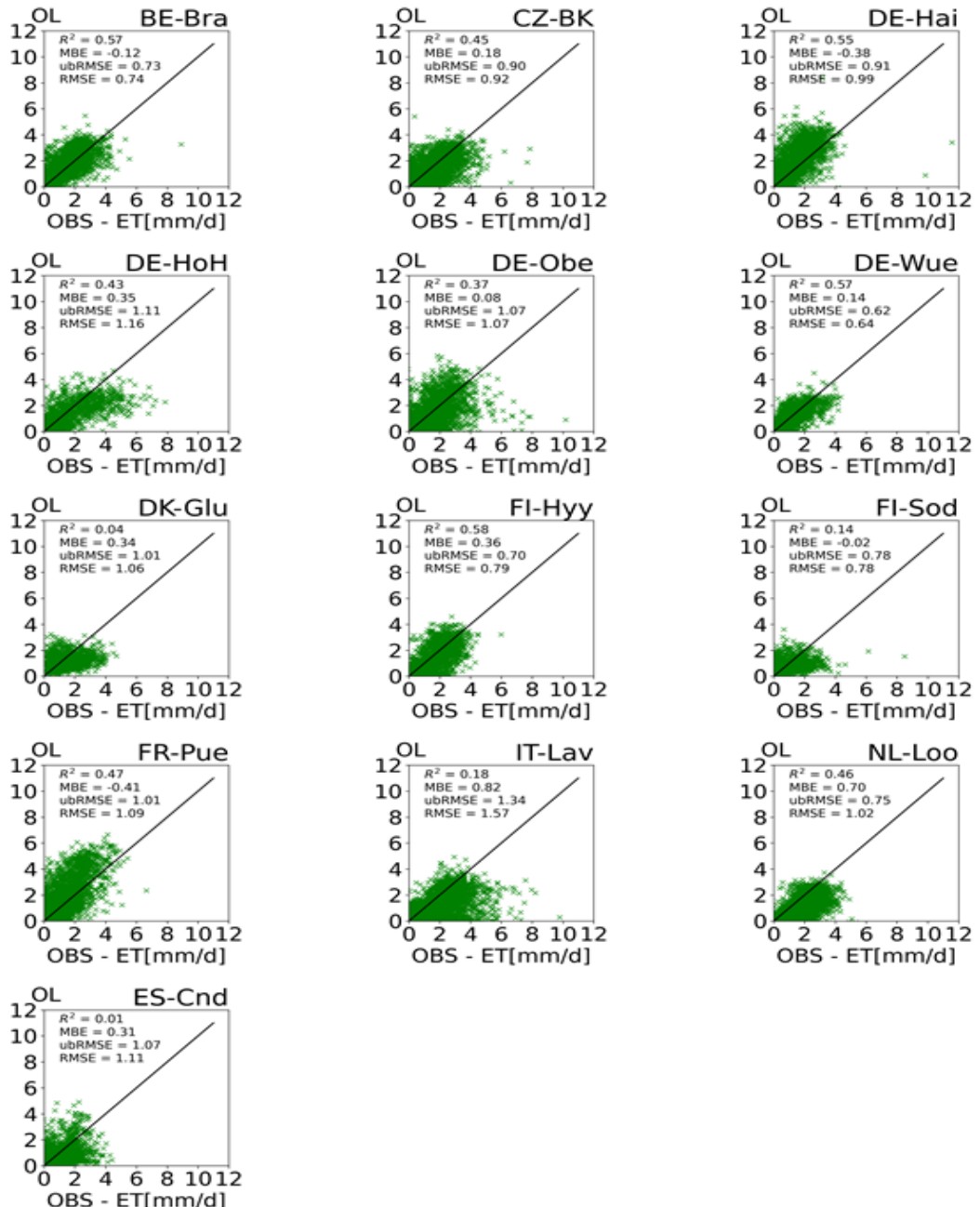

**Figure 10**: Scatter plots of observed evapotranspiration at thirteen study sites versus OL simulation results. The points represent daily averages for the days on which observation data are available.

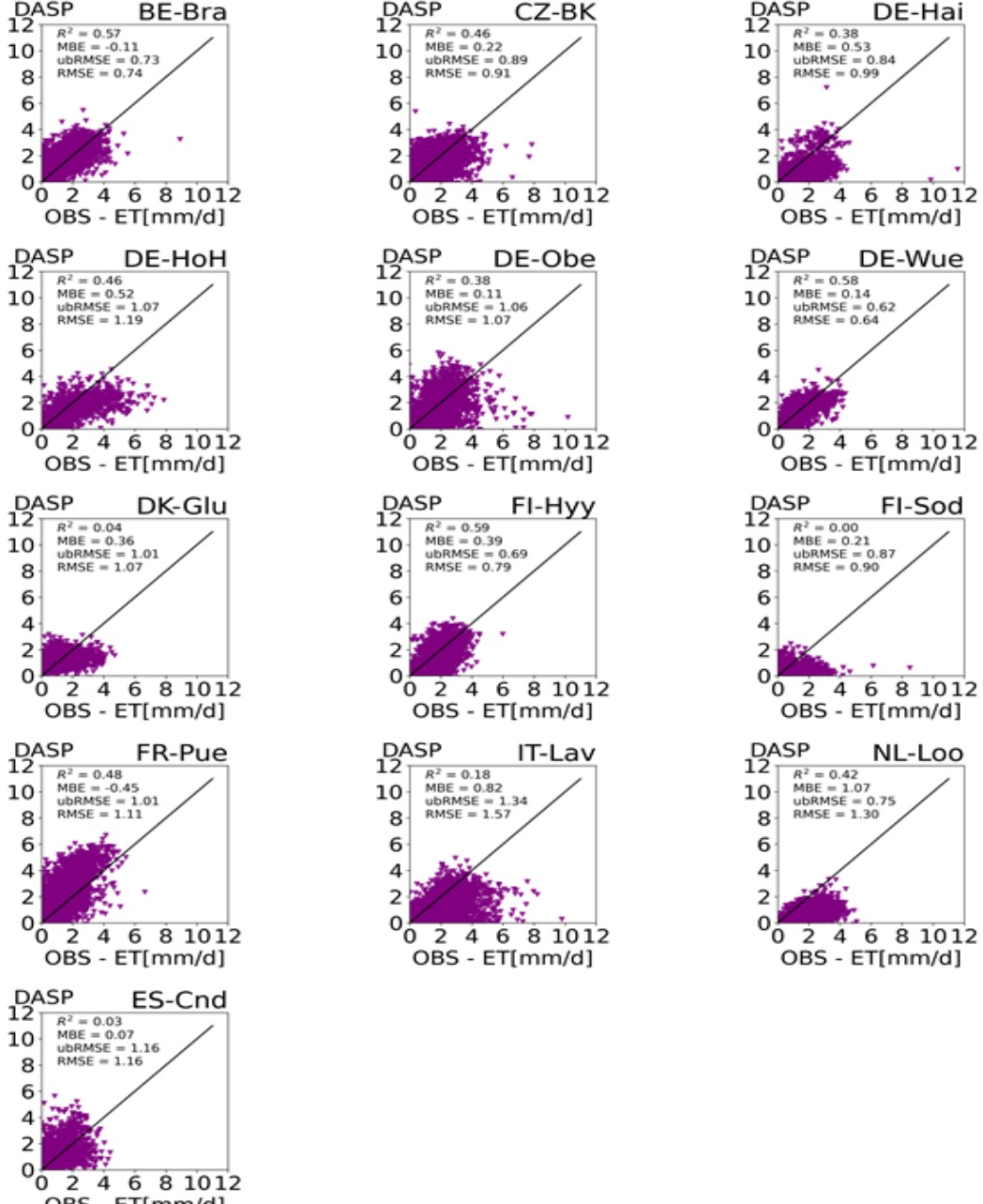

**Figure 11:** Scatter plots of observed evapotranspiration at thirteen study sites versus DASP simulation results. The points represent daily averages for the days on which observation data are available.

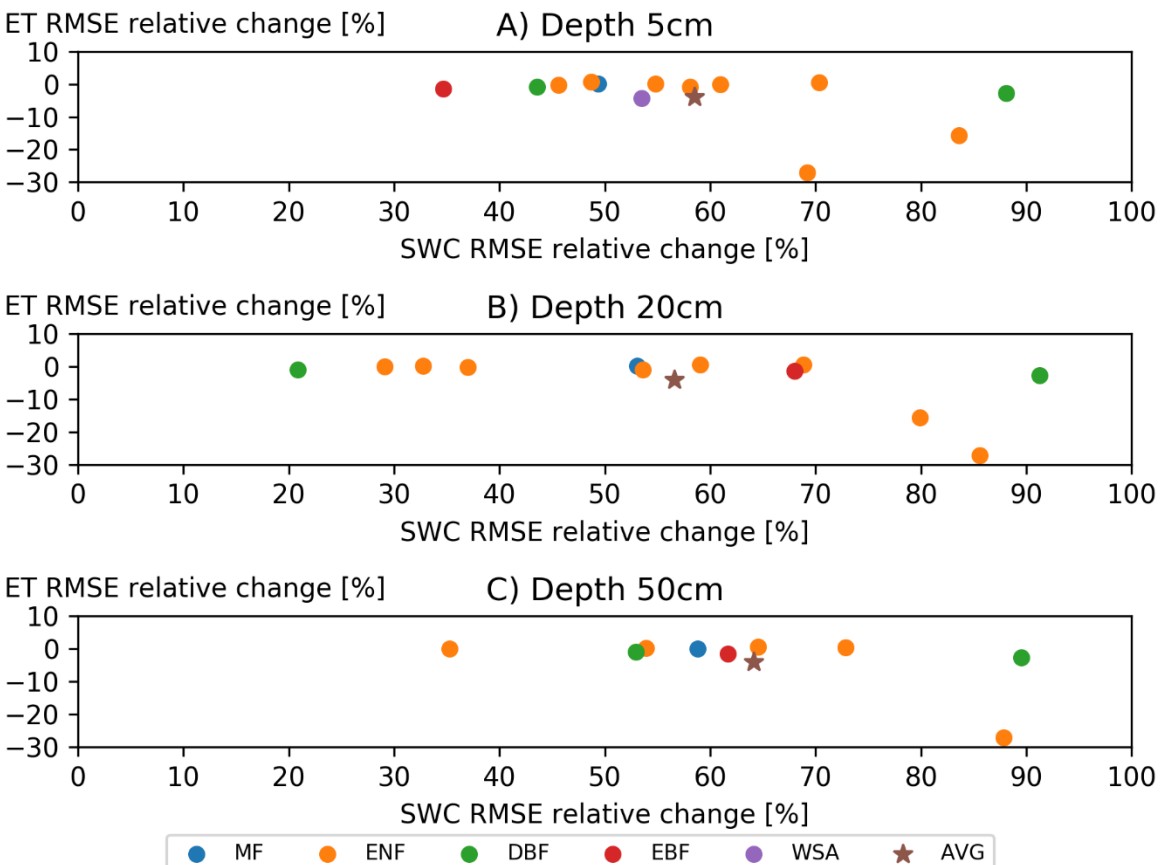

**Figure 12:** Comparing the SWC and ET characterization for the OL and DASP simulations. Each point represents the overall average RMSE change for one site. The color of the points indicates the classification code for the different forest types (MF: mixed forest, ENF: evergreen needle leaf forest, DBF: deciduous broad leaf forest, EBF: evergreen broad leaf forest, AVG: average over all forest types).

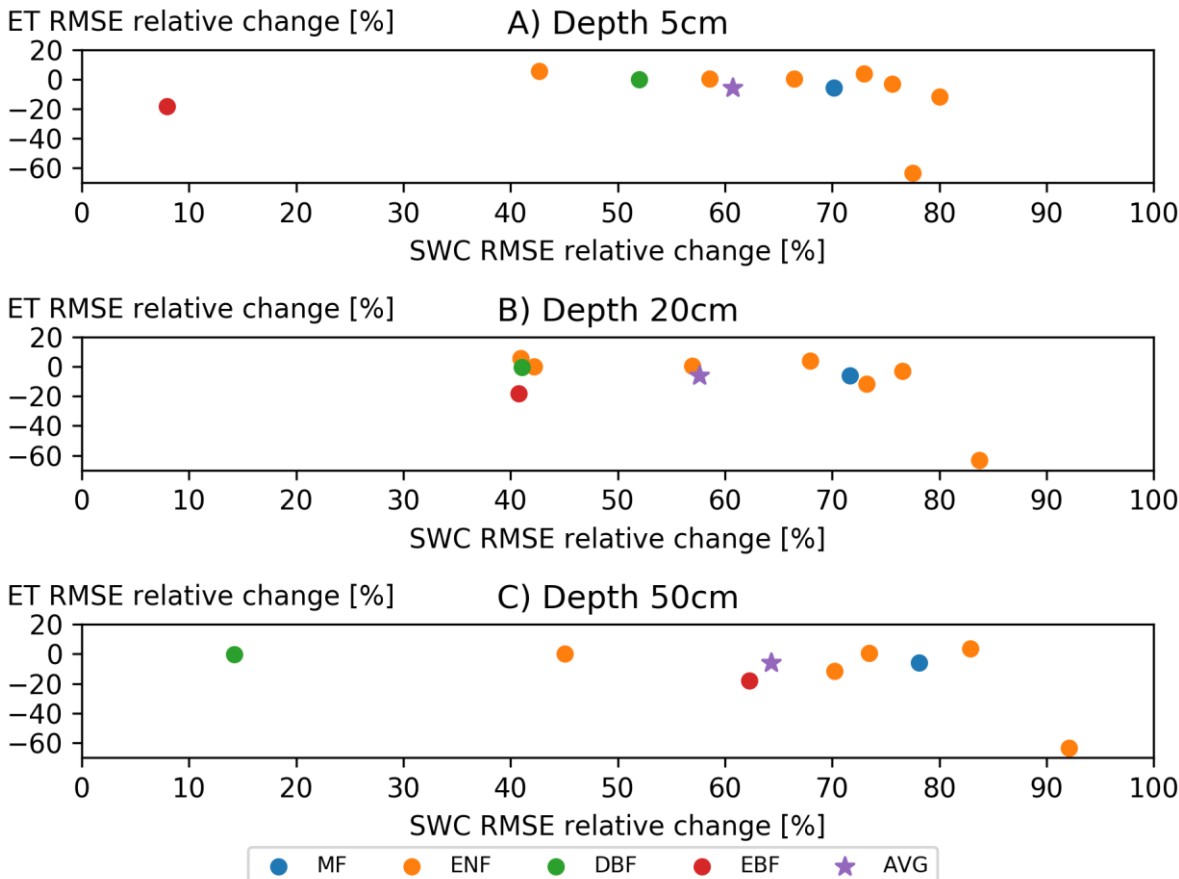

**Figure 13**: Comparing the SWC and ET characterization for the OL and DASP simulations using CLM5-SP. Each point represents the overall average RMSE change for one site. The color of the points indicate the forest type (MF: mixed forest, ENF: evergreen needle leaf forest, DBF: deciduous broad leaf forest, EBF: evergreen broad leaf forest, AVG: average over all forest types).

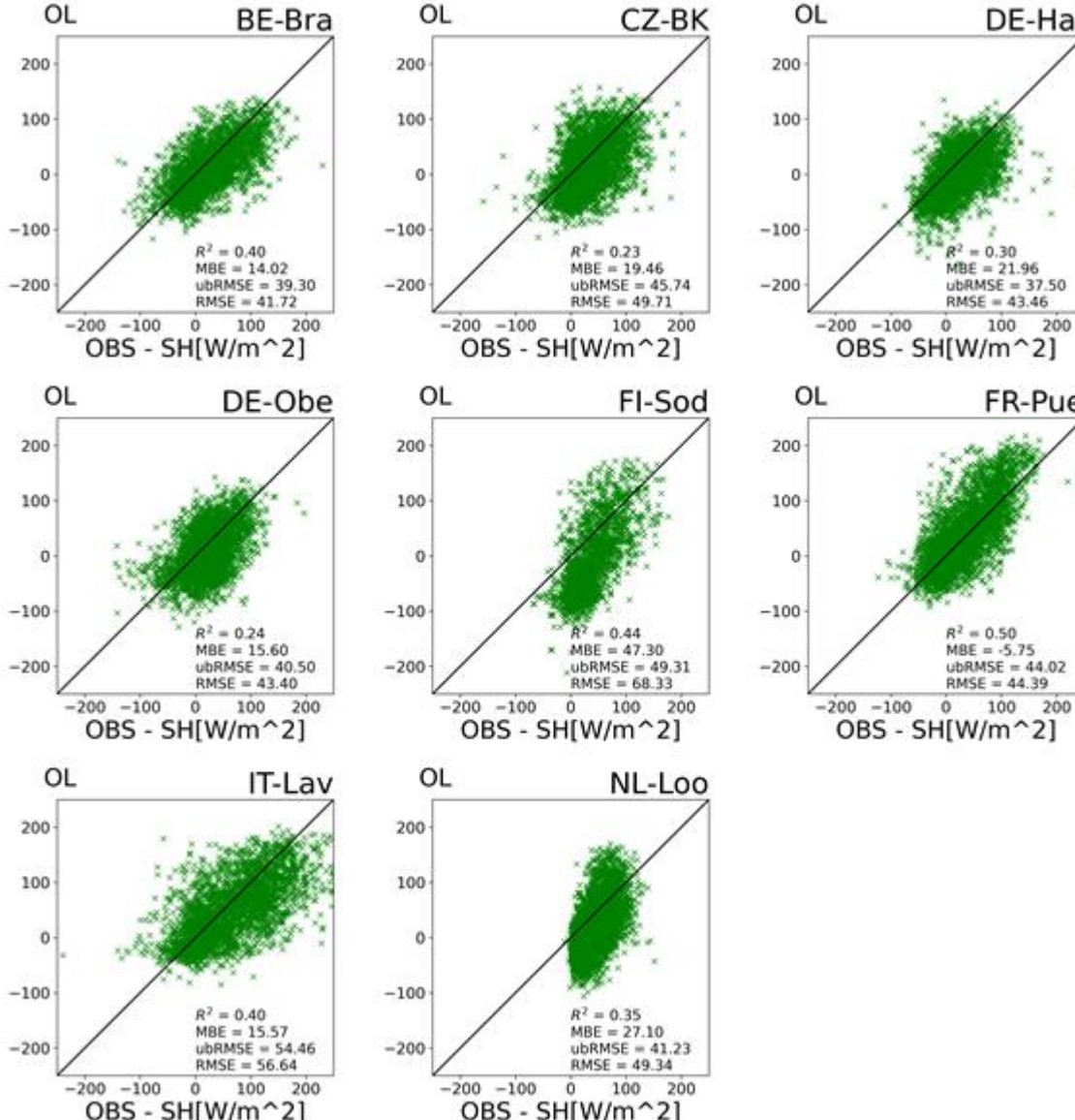

**Figure 14:** Scatter plots of observed sensible heat flux at eight study sites versus OL simulation results. The points represent daily averages for the days on which observation data are available.

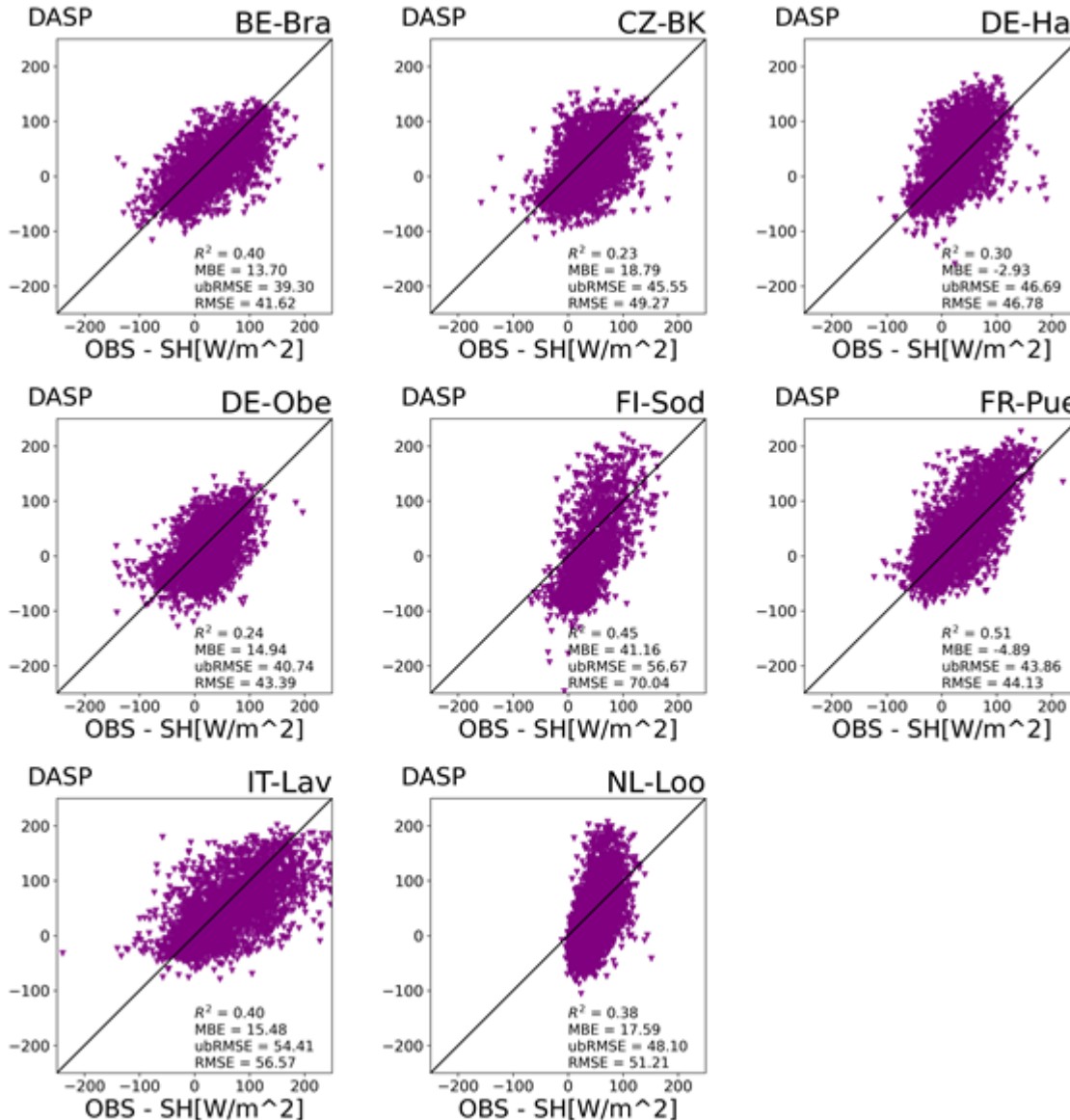

**Figure 15**: Scatter plots of observed sensible heat flux at eight study sites versus DASP simulation results. The points represent daily averages for the days on which observation data are available.

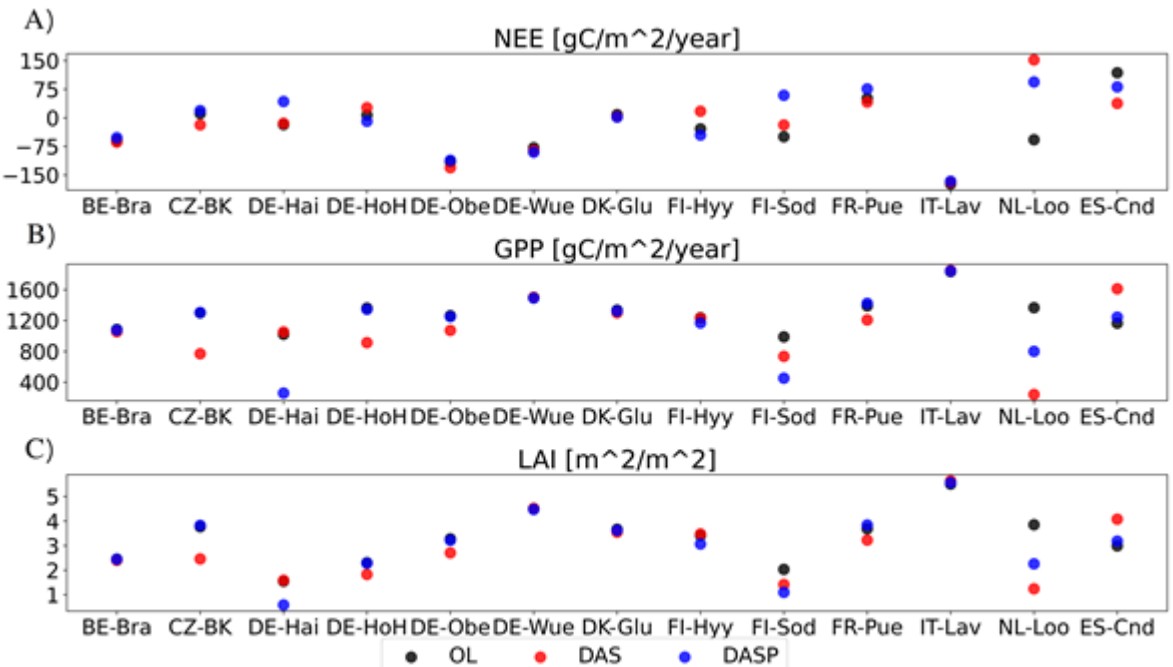

**Figure 16**: Open loop (OL) and assimilation scenario (DAS and DASP) yearly averages of A) net ecosystem exchange (NEE), B) gross primary production (GPP), and C) leaf area index (LAI) for all selected sites.

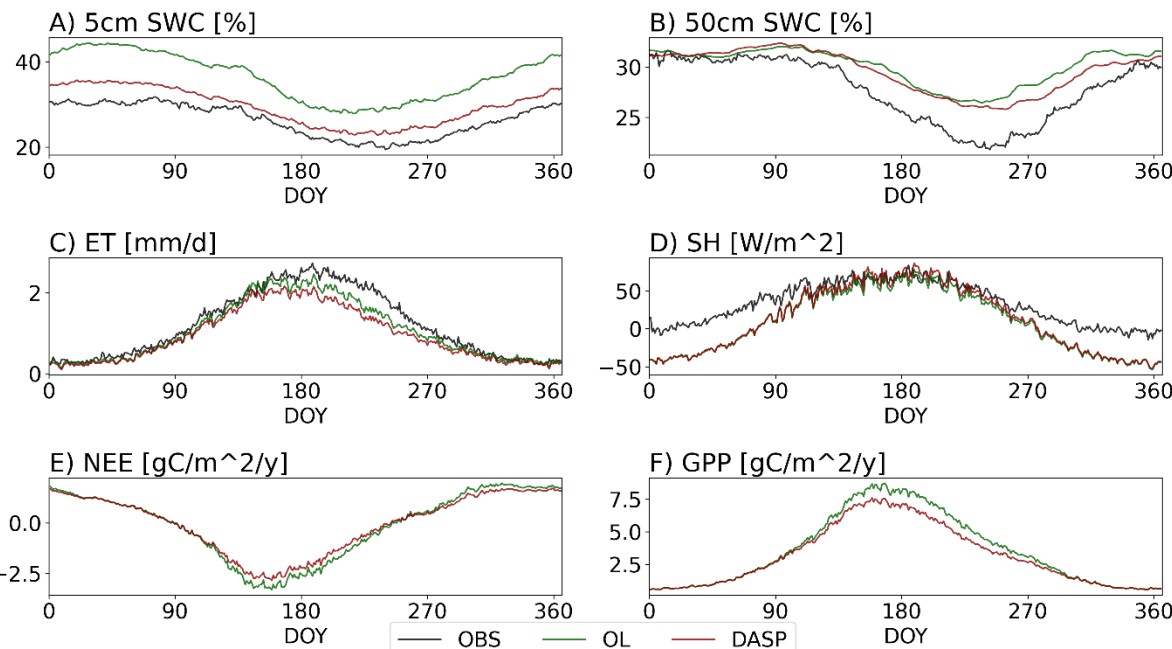

**Figure 17:** Seasonality of observed (OBS) and simulated (OL and DASP) states and fluxes based on daily averages from all years (2009 to 2018) and all sites: A) soil water content (SWC) at 5 cm depth, B) SWC at 50 cm depth, C) evapotranspiration (ET), D) sensible heat flux (SH), E) net ecosystem exchange (NEE), and F) gross primary production (GPP).

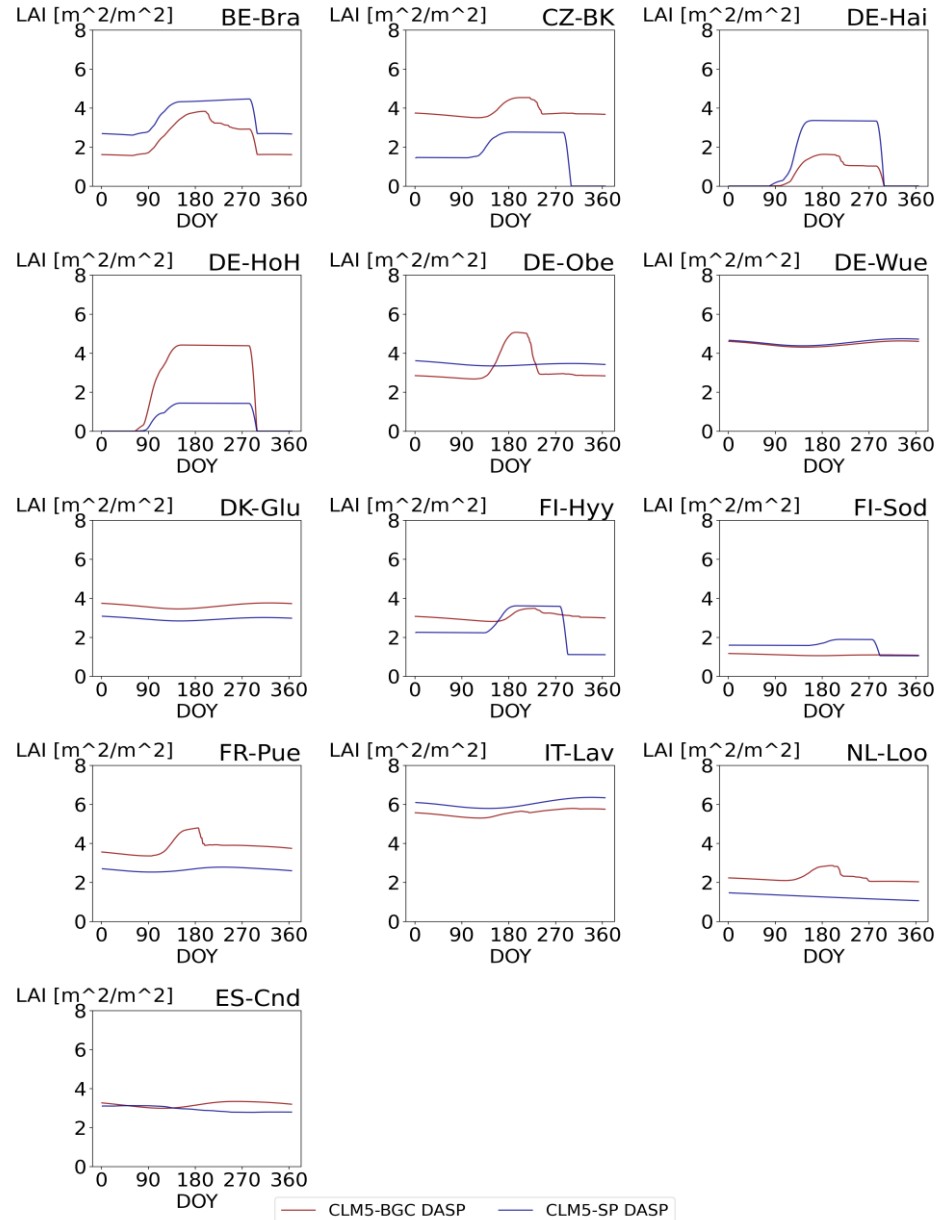

**Figure 18:** Seasonality of simulated LAI for each of the sites. The red line represents predicted LAI from the CLM5-BGC-DASP simulations and the blue line represents LAI inputs used in the CLM5-SP-DASP simulations.

# Appendix.

## A. Additional figures

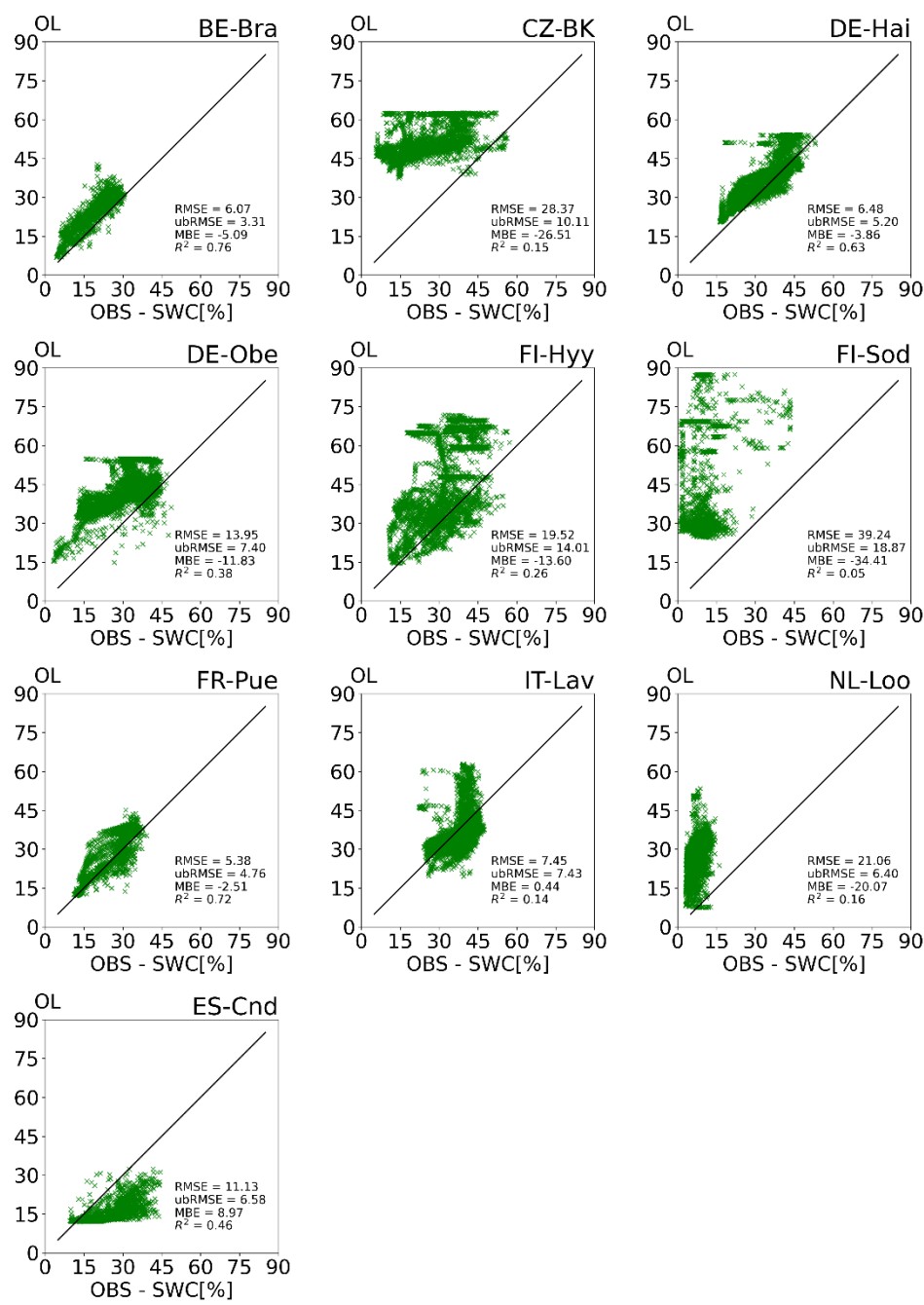

**Figure A1** : Scatter plots of observed soil water content at ten study sites versus OL simulation results at

5 cm depth. The points represent daily averages for the days on which observation data are available.

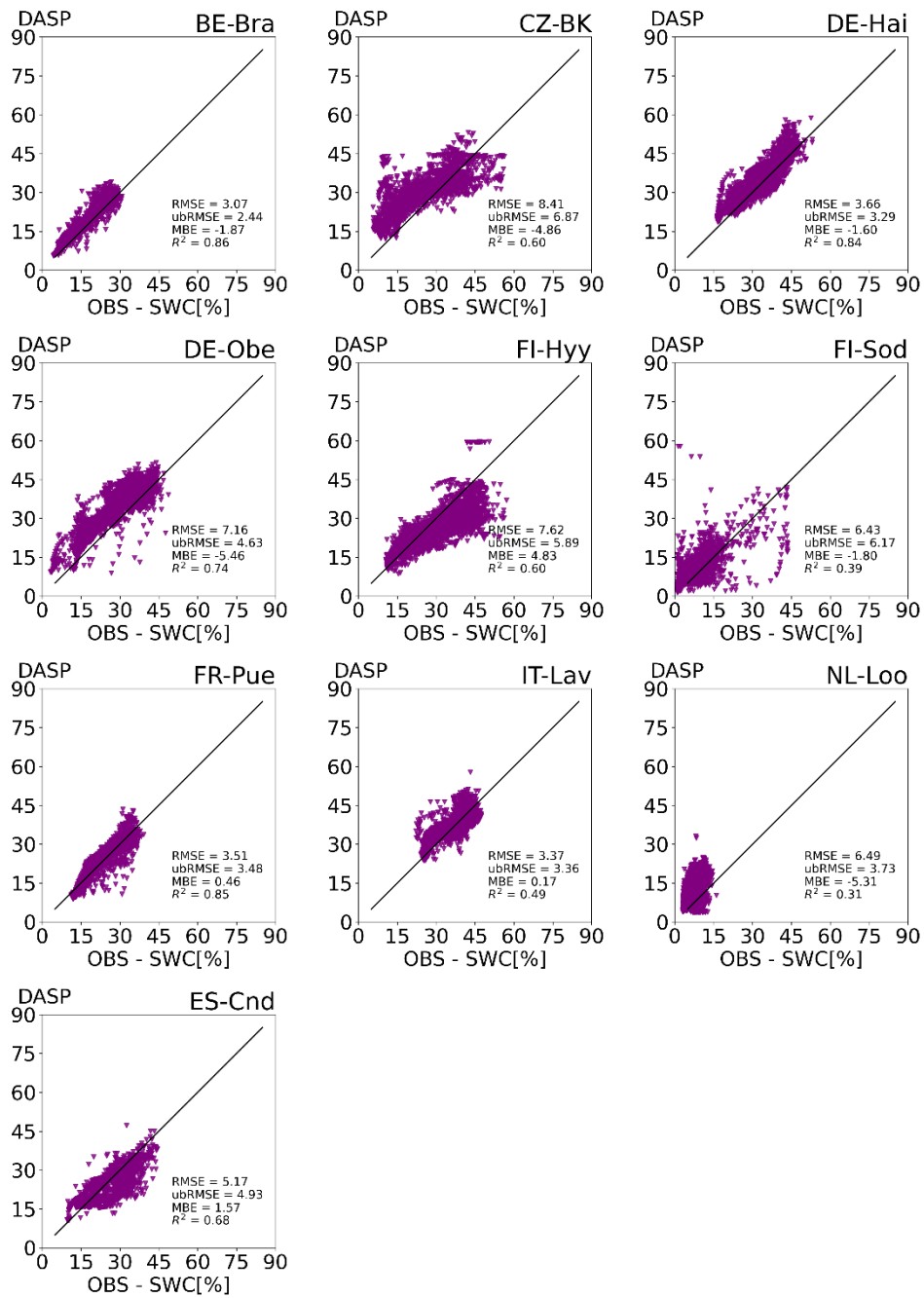

**Figure A2** Scatter plots of observed soil water content at ten study sites versus DASP simulation results at 5 cm depth. The points represent daily averages for the days on which observation data are available.

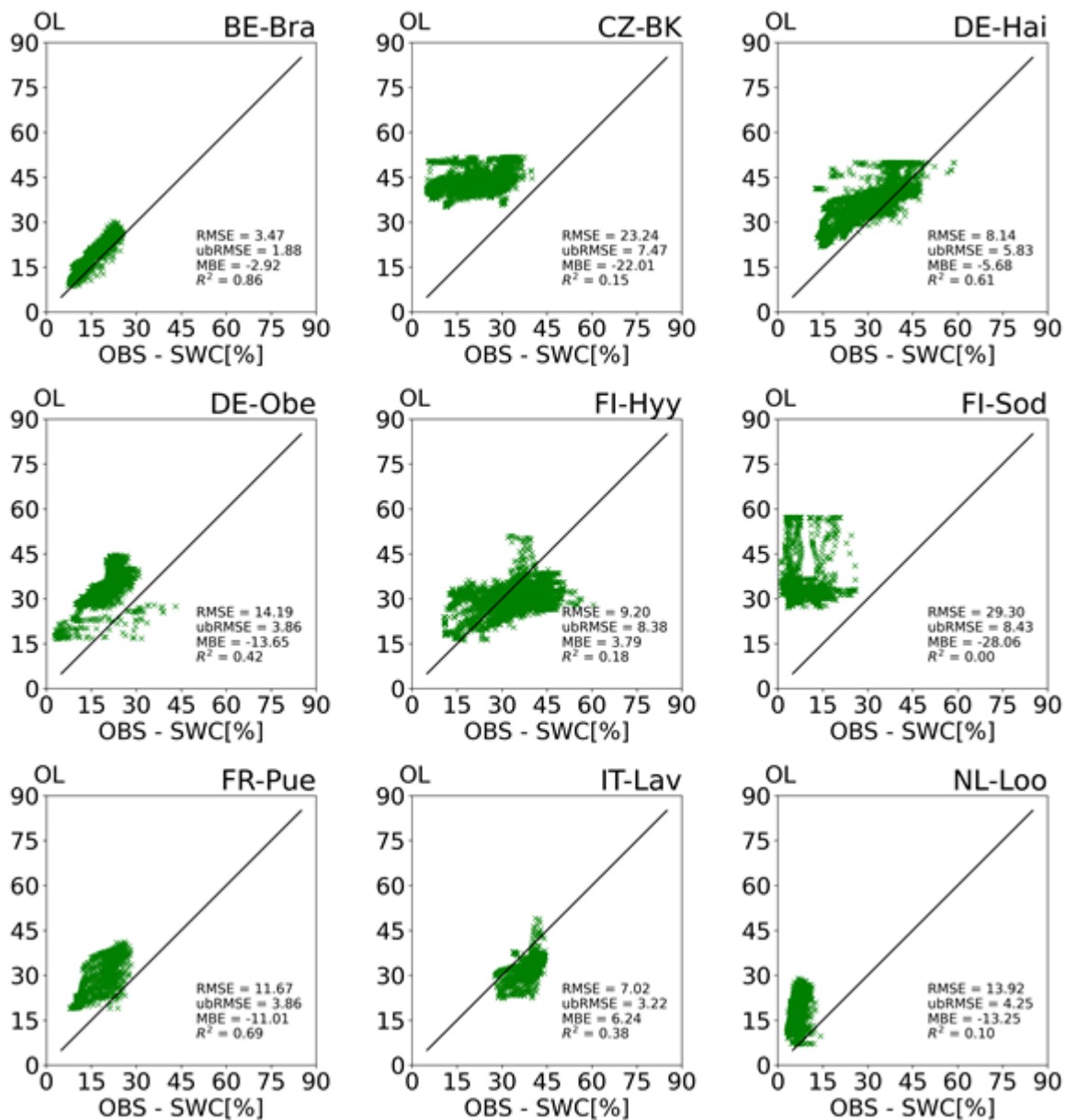

**Figure A3** : Scatter plots of observed soil water content at ten study sites versus OL simulation results at 20 cm depth. The points represent daily averages for the days on which observation data are available.

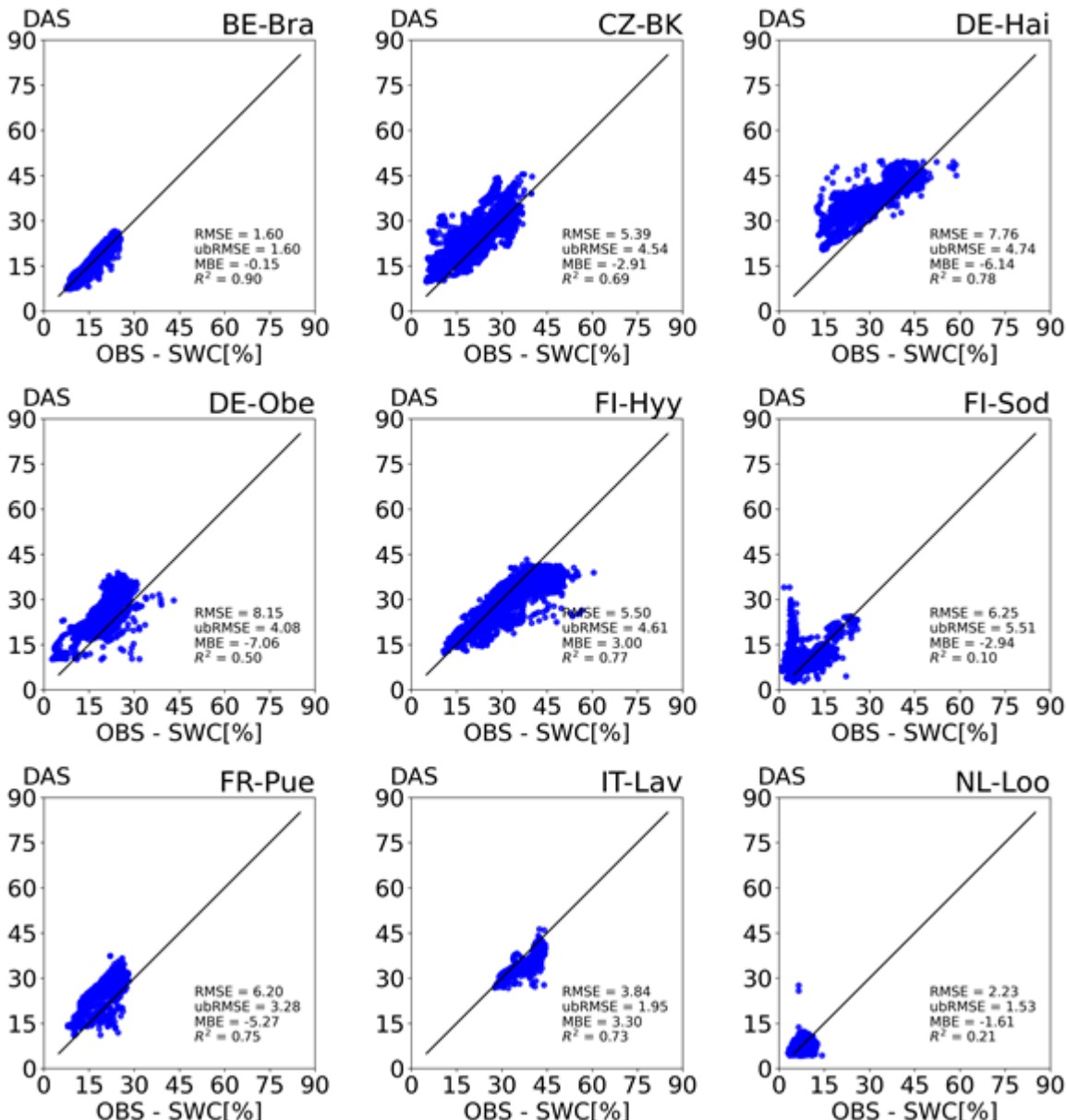

**Figure A4** : Scatter plots of observed soil water content at ten study sites versus DAS simulation results at 20 cm depth. The points represent daily averages for the days on which observation data are available.

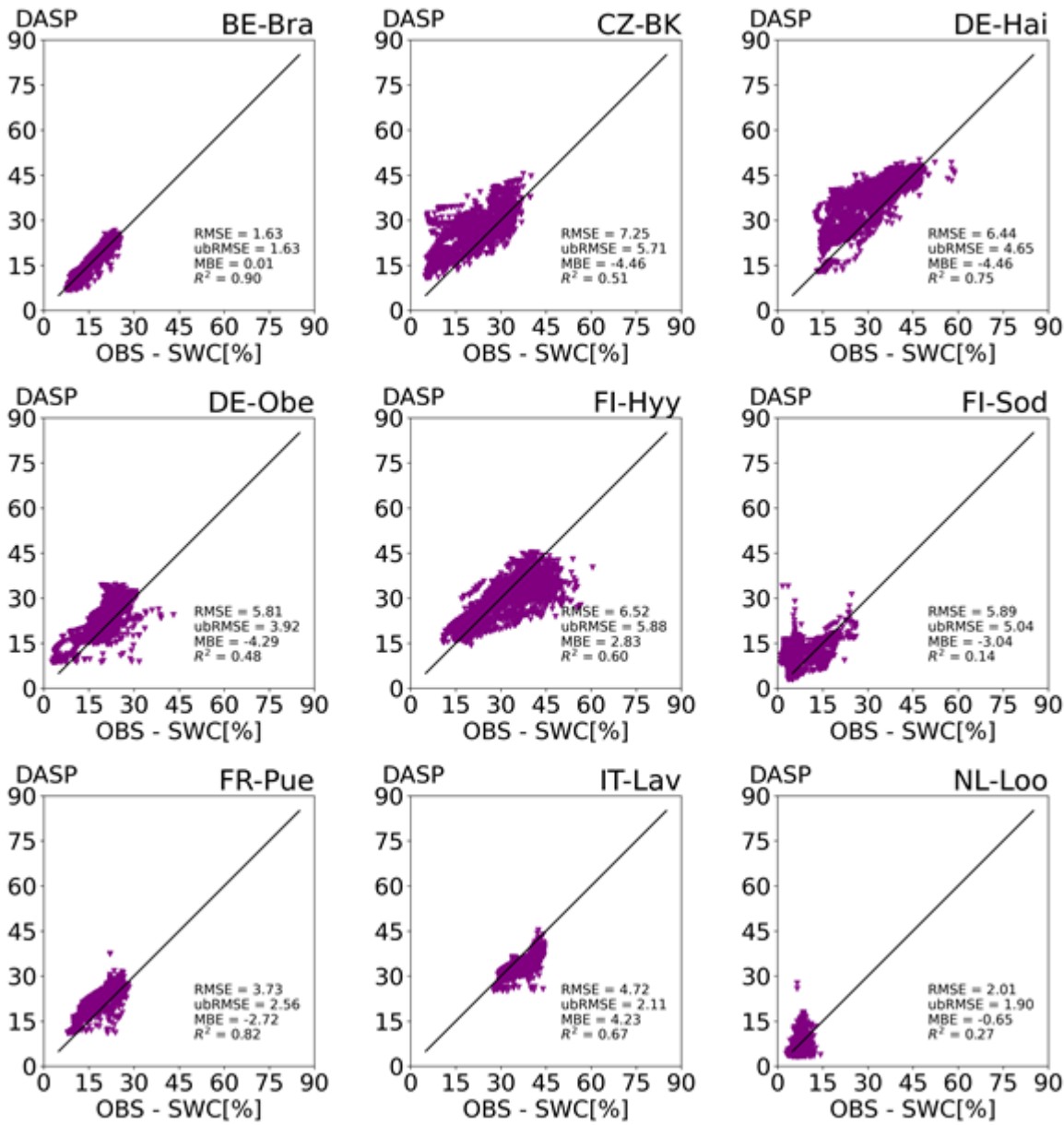

**Figure A5** : Scatter plots of observed soil water content at ten study sites versus DASP simulation results at 20 cm depth. The points represent daily averages for the days on which observation data are available.

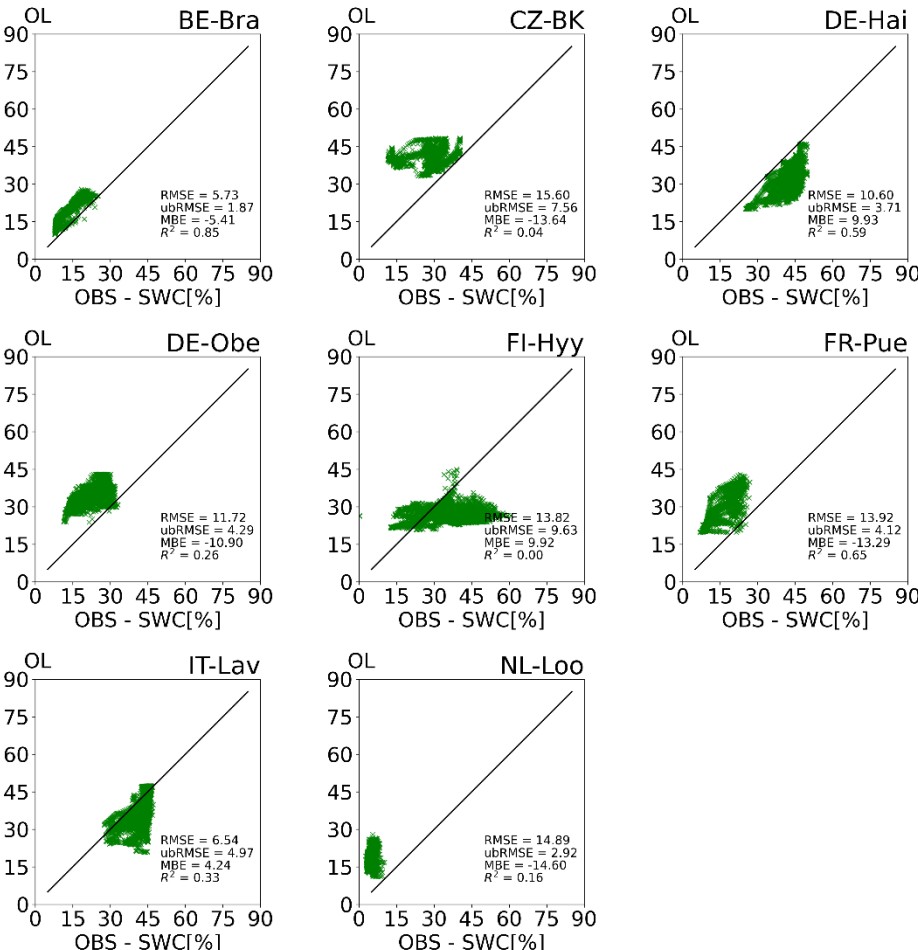

**Figure A6**: Scatter plots of observed soil water content at eight study sites versus OL simulation results at 50 cm depth. The points represent daily averages for the days on which observation data are available.

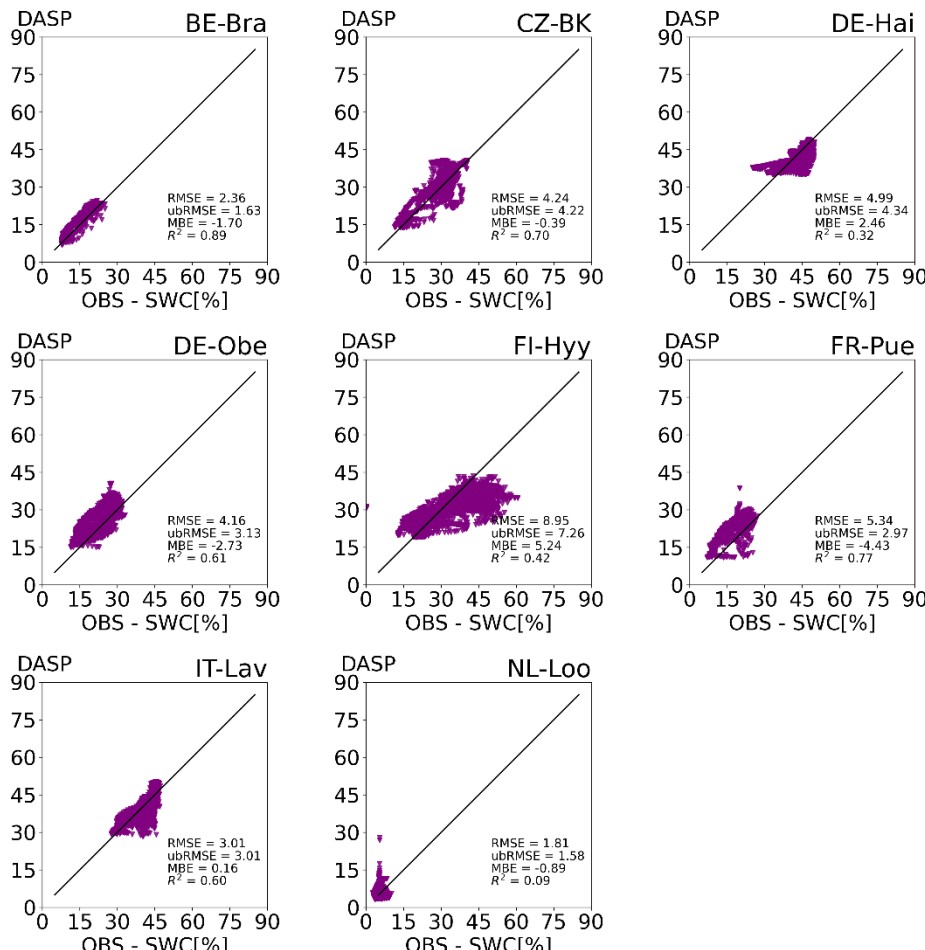

**Figure A7**: Scatter plots of observed soil water content at eight study sites versus DASP simulation results
at 50 cm depth. The points represent daily averages for the days on which observation data are available.

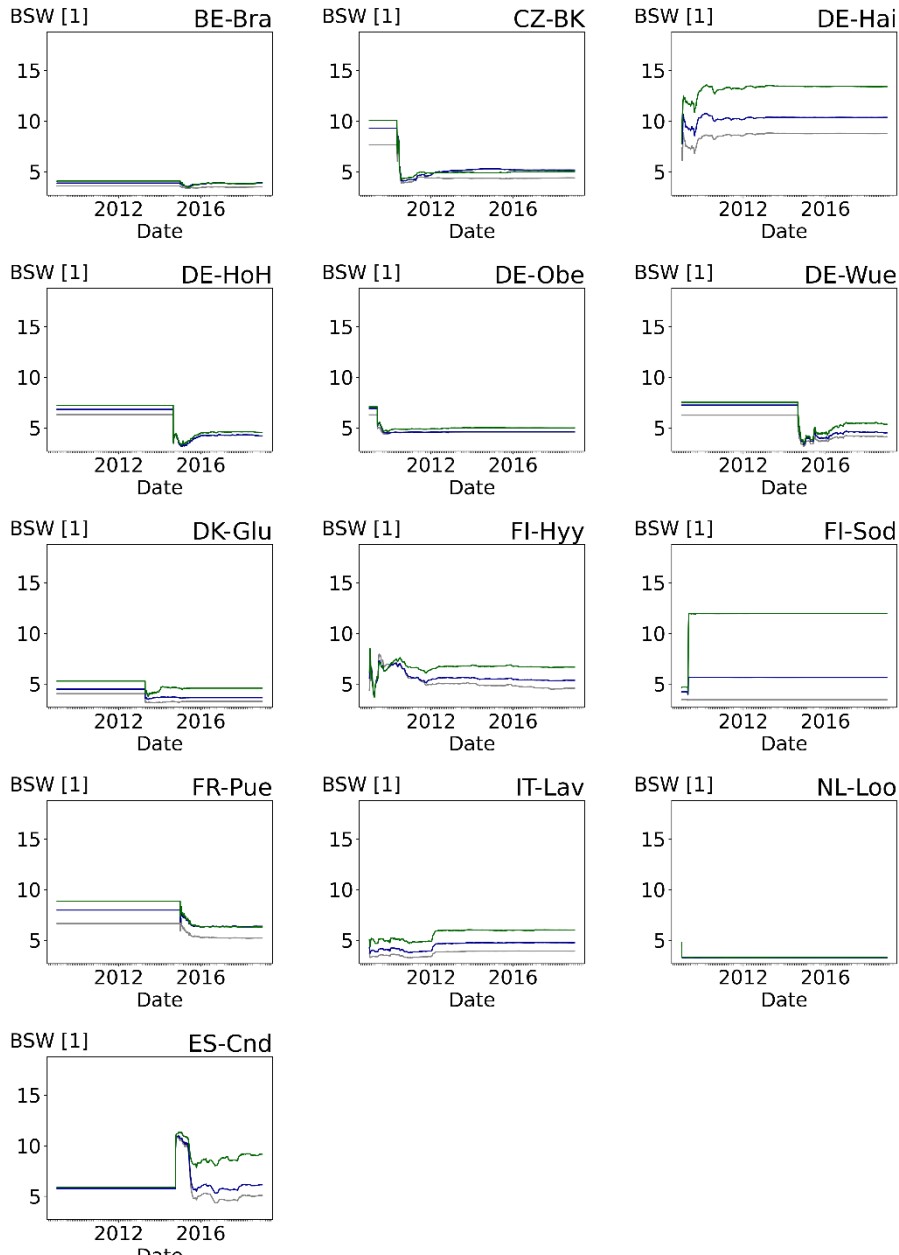

**Figure A8**: Time series .for the Clapp-Hornberger shape parameter B (BSW) for each site in the DASP simulation. The grey line is the value at 5 cm depth, the blue line at 20cm depth, and the green line at 50cm depth.

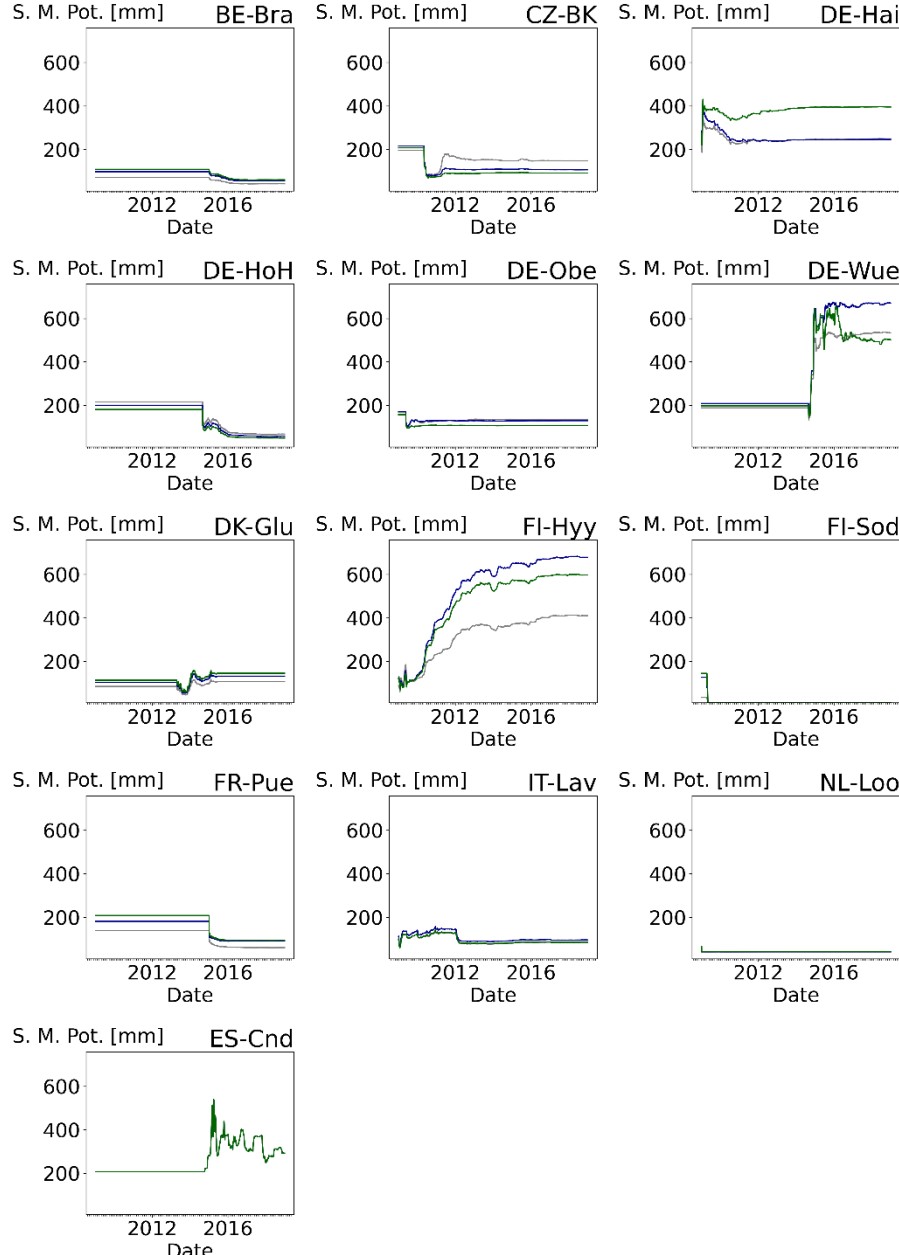

**Figure A9**: Time series .for the saturated soil matric potential for each site in the DASP simulation. The grey line is the value at 5 cm depth, the blue line at 20cm depth, and the green line at 50cm depth.

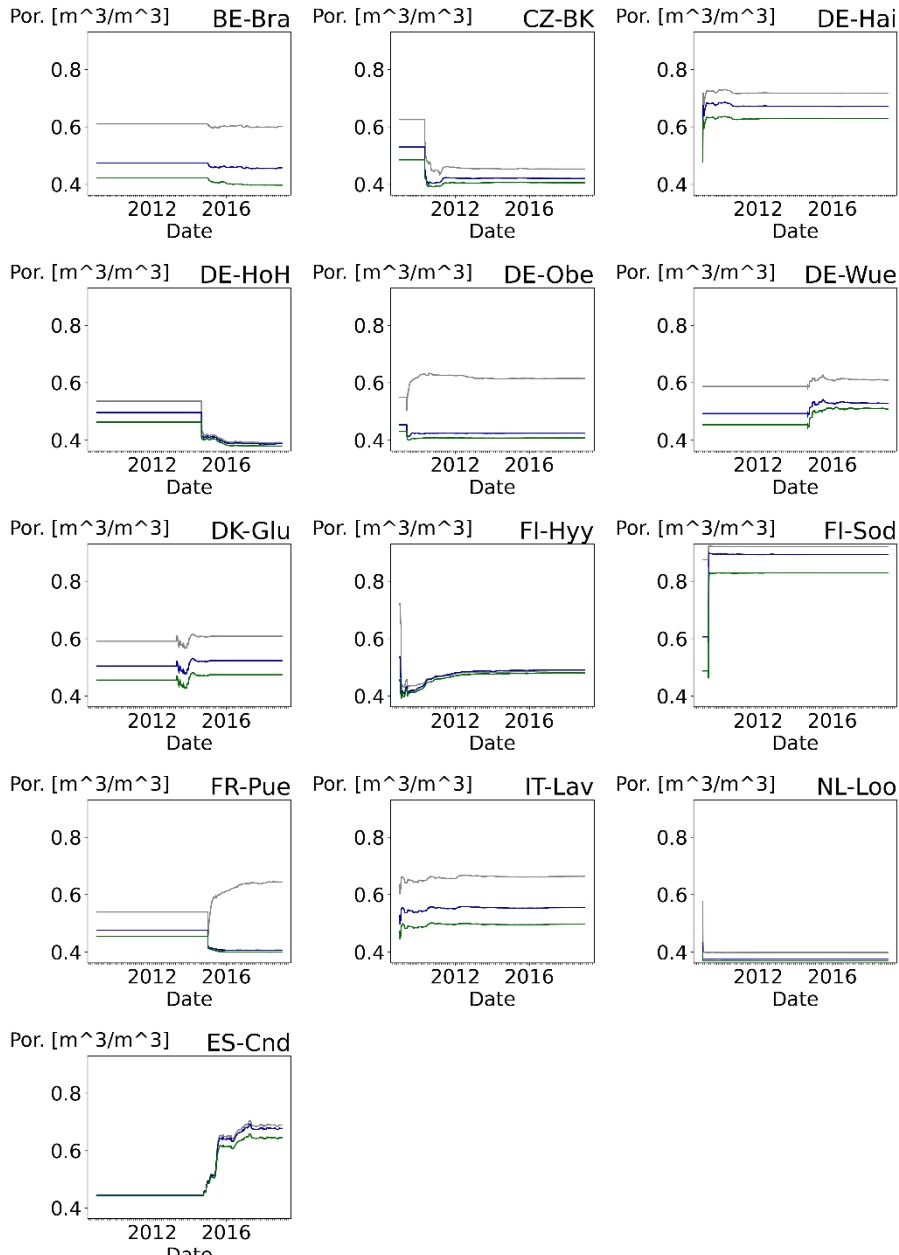

**Figure A10**: Time series .for the porosity for each site in the DASP simulation. The grey line is the value at 5 cm depth, the blue line at 20cm depth, and the green line at 50cm depth.

**Code availability.**

The code used in this study is available at https://github.com/HPSCTerrSys/TSMP .

**Data availability.**

Data for the European FLUXNET sites is available at http://www.europe-fluxdata.eu/. Some additional data used in this study is from the eLTER data portal ( https://data.lter-europe.net/) and ICOS data portal (https://www.icos-cp.eu/). The CRNS data is published in "Dataset of COSMOS-Europe: A European network of Cosmic-Ray Neutron Soil Moisture Sensors" available at https://doi.org/10.34731/x9s3-kr48 (Bogena and Ney, 2021).

**Competing interests.** The authors declare the following competing interests: Harrie-Jan Hendricks Franssen is a part of the editorial board of Hydrology and Earth System Sciences.

**Author contribution.**

L.S. pre-processed the data, adjusted the code, performed the simulations, and prepared the manuscript. M.A. and S.A-B. provided data and contributed to the manuscript. H.B., H.J.H.F., and H.V. supervised the research, co-designed the experiments, and contributed to the manuscript.

**Acknowledgements**.

The authors gratefully acknowledge the support by the project "LIFE RESILIENT FORESTS – Coupling water, fire and climate resilience with biomass production from forestry to adapt watersheds to climate change". Additionally, the authors gratefully acknowledge the support by the Deutsche Forschungsgemeinschaft (DFG, German Research Foundation) – SFB 1502/1–2022 - Projekt-nummer: 450058266. Furthermore, the authors gratefully acknowledge the computing time granted through JARA on the supercomputer JURECA at Forschungszentrum Jülich. We are thankful for all the data provided by FLUXNET, eLTER, eLTER PLUS, ICOS, COSMOS-Europe projects and we thank all the site PI and technical staff of the sites shown in this study.