# Peer review of "Evapotranspiration prediction for European forest sites does not improve with assimilation of insitu soil water content data"

_EGUsphere, 2023_

## Author Response (AR1)

**Scientific/Detailed Comments:** Referee comments in bold and author answer non-bold.

**Review comments by Referee #1**

**In this manuscript the authors described assimilation of soil moisture observations into the land surface model CLM5. In general, the paper is well structured, described and written. I have several questions which are still unclear for me.**
**1) as you mentioned in the end of section 2.1, you leave the data gaps. I'm wondering your observation frequency.**

The observation frequency varies among data sources, e.g. FLUXNET provides data in 30-minute intervals. However, as mentioned earlier in section 2.1, we use daily averages of the soil water content, evapotranspiration, and sensible heat for the assimilation and analysis.

Changes in text:

Lines: 134 - 139

"We only assimilated (daily mean averaged) soil water content observations when measurements were available for a given day. The daily mean averages were calculated independent from the observation frequency for the different sites. Similarly, simulated evapotranspiration was only compared with observations when data were available, on the basis of daily mean averages."

**2) for parameter updating in 2.3.2, you update the solid fractions instead of hydraulic conductivity. Please explain the reason. Does it work better, or what's the particular reason for this?**

We performed data assimilation using a direct approach to update combinations of soil saturated hydraulic conductivity, hydraulic conductivity exponent, porosity, and soil matric potential but found that indirectly updating sand, clay, and organic matter fractions using pedotransfer functions provides better results. Specifically, we found that parameters often approached upper or lower defined limits, which did not happen with the indirect approach. Updating the solid fractions provided in general more stable parameter estimates and better simulation results.

Changes in text:

Lines: 220 - 230

"In previous studies parameters were updated indirectly (Naz et al., 2019; Han et al., 2014; Baatz et al., 2017). We tested directly updating saturated hydraulic conductivity, porosity, hydraulic conductivity exponent, and soil matric potential but this resulted in more unstable estimates than indirectly updating soil hydraulic parameters. The pedotransfer function which is used for the indirect updating results in reasonably correlated soil hydraulic parameters. In this study, the parameters are chosen to optimize the SWC estimation and not ET estimation to study the effects of SWC improvements on ET. To more directly improve the ET estimation, parameters affecting the ET process directly should be added, e.g. vegetation hydraulic parameters. "

**3) line 175: how about the observation errors vs. perturbations?**

**(Line 175: "Only the PFT were manually assigned for each site. For the ensemble creation, the fractions ofsand, clay, and organic matter are modified for each ensemble member. The perturbations are normally distributed with mean zero and a standard deviation of 10%.")**

For the data assimilation the observation error is assumed be constant and set to a root-mean square error of 0.02 cm³/cm³.

Changes in text:
Lines: 201 - 203

"In this study, the state vector depends on the simulation scenario (explained in more detail in section 2.3.2) and R is based on the measurement errors which are assumed to be constant and independent with a root-mean square error of 0.02 cm³/cm³."

**4) line 190: parameter updating refers to fractions of sand, clay and organic? Are there any other parameters included, e.g. the vG parameters and porsity? If not, why do you only update these parameters?**

Yes, parameter updating refers to the sand, clay, and organic matter fractions. We use the indirect approach in which the soil hydraulic parameters are calculated from these characteristics using the Clapp and Hornberger (1978) pedotransfer function as mentioned in 2.3.2. CLM5 uses the Brooks-Corey parameters and not the vG parameters as mentioned in section 2.2. As already mentioned in comment 2), we found that the indirect approach via pedotransfer functions provides better results than direct updating of hydraulic parameters.

Changes in text:
Lines: 220 - 230

"In previous studies parameters were updated indirectly (Naz et al., 2019; Han et al., 2014; Baatz et al., 2017). We tested directly updating saturated hydraulic conductivity, porosity, hydraulic conductivity exponent, and soil matric potential but this resulted in more unstable estimates than indirectly updating soil hydraulic parameters. The pedotransfer function which is used for the indirect updating results in reasonably correlated soil hydraulic parameters. In this study, the parameters are chosen to optimize the SWC estimation and not ET estimation to study the effects of SWC improvements on ET. To more directly improve the ET estimation, parameters affecting the ET process directly should be added, e.g. vegetation hydraulic parameters."

**Review comments by Referee #2**

**This reviewer had two critical concerns about this manuscript related to the simulation of leaf area index, and the design and implementation of the parameter estimation:**

**First, I don't see how the author's came to the conclusion "[the] results suggest that state-of-the-art LSM such as CLM5 still suffer from uncertainties in the representation of soil hydrological processes in forests, e.g. deep root water uptake, uncertainties in the representation of biological processes of tree transpiration", without accoutning for biases of the simulated leaf area for the sites. Running CLM5-BGC can lead to erroneous leaf area values, and is why CLM users often use CLM-SP (prescribed leaf area) to diagnose to what extent the simulation of leaf area is impacting your soil moisture and ET relationship. Have a look at Li et al., (2022) or Fox et al., (2022) as an example of how prescribing or assimilating observations of leaf area can effect the representation of carbon and water cycling. https://agupubs.onlinelibrary.wiley.com/doi/10.1029/2021MS002747**

**If the simulated leaf area is incorrect in magnitude and timing it is unlikely that improving the soil water content through DA will improve ET. The authors do not account for this in the manuscript.**

We agree that running CLM5-BGC can lead to erroneous LAI simulation results and that CLM-SP can be used to diagnose to what extent this uncertainty affects the simulation of soil moisture and ET. Therefore, we now also explore the difference between CLM-SP and CLM-BGC simulations and have included the corresponding results in the revised manuscript.

Changes in text:

Lines: 24 – 29, 303 - 313

"This finding indicates that only improving the SWC estimation of state of the art LSM such as CLM5 is not sufficient to improve evapotranspiration estimates for forest sites. To improve evapotranspiration estimates, it is also necessary to consider the representation of LAI in magnitude and timing, as well as uncertainties in water uptake by roots and vegetation parameters."

"This could be caused by the mismatch of simulated and actual LAI for these sites. To investigate this, we repeated the simulations using CLM5 with satellite-derived phenology (CLM5-SP). The results are shown in Fig. 9. For CLM5-SP we observe an average improvement in the RMSE of SWC between 57.6 % and 64.3 % and an average RMSE deterioration of 5.8 % for the ET estimation. These CLM5-SP simulations use the default datasets from CLM5 and without site specific calibration of the timing or magnitude of the seasonal phenology of LAI. Therefore, even for the CLM5-SP simulations, there is a mismatch between simulated and actual LAI. However, also for this case, there are sites with a large improvement in SWC estimation that show deterioration in the ET estimation."

Another possible explanation for the improvement in SWC estimation but no improvement of ET estimation is the underestimation of root water uptake from deeper soil layers for forest sites, as also suggested by Shrestha et al. (2018). "

[Figure]

Figure 9: Comparing the SWC and ET characterization for the OL and DASP simulations using CLM5-SP. Each point represents the overall average RMSE change for one site. The color of the points indicate the forest type (MF: mixed forest, ENF: evergreen needle leaf forest, DBF: deciduous broad leaf forest, EBF: evergreen broad leaf forest, AVG: average over all forest types).

**Second, in reference to the parameter estimation, the authors update the soil layer fractions sand/clay/organic which then impact to what extent the true soil hydraulic properties are weighted in the soil layer.  It is a strange approach to access the 'true' global hydraulic parameters through a fixed land surface property.  This would be equivalent to treating the PFT for each site as a 'parameter' rather than a fixed/prescribed property, and varying the PFT fraction across MF/ENF/DBF/ENF/WSA for each site, which would then weight the parameter set associated with each PFT.  This is not done, and the authors don't do that here. Instead the author's prescribe the PFT for each site and keep that fixed.  Similarly, it is unclear why the author's would then not prescribe the sand/clay/organic layers either from the default CLM files, or prescribe the sand/clay/organic fractions from the FLUXNET site data.**

Soil texture data are always subject to uncertainties and therefore it is reasonable to not fix them. Therefore, the indirect approach of updating soil characteristics, i.e. sand, clay, and organic matter fraction, to update the soil hydraulic parameters using a pedotransfer function was used in various studies (Naz et al., 2019; Han et al., 2014; Baatz et al. 2017).

Secondly, we performed data assimilation using a direct approach to update combinations of soil saturated hydraulic conductivity, hydraulic conductivity exponent, porosity, and soil matric potential but found that indirectly updating sand, clay, and organic matter fractions using pedotransfer functions provides better results. Specifically, we found that parameters often approached upper or lower defined limits, which did not happen with the indirect approach. Updating the solid fractions provided in general more stable parameter estimates and better simulation results.

Changes in text:

Lines: 220 – 230

"In previous studies parameters were updated indirectly (Naz et al., 2019; Han et al., 2014; Baatz et al., 2017). We tested directly updating saturated hydraulic conductivity, porosity, hydraulic conductivity exponent, and soil matric potential but this resulted in more unstable estimates than indirectly updating soil hydraulic parameters. The pedotransfer function which is used for the indirect updating results in reasonably correlated soil hydraulic parameters. In this study, the parameters are chosen to optimize the SWC estimation and not ET estimation to study the effects of SWC improvements on ET. To more directly improve the ET estimation, parameters affecting the ET process directly should be added, e.g. vegetation hydraulic parameters."

**Furthermore, the choice of parameters to be estimated should more closely relate to processes controlling the SWC-ET relationship, such as stomatal conductance, and the \*vegetation\* hydraulic parameters specifically involved in the Plant Hydraulic Stress formulation in CLM5. The soil hydraulic parameters estimated in the manuscript, seem to control the dynamics of moisture within the soil layer, but not the transfer of water from root to leaf, which is controlled by the stomatal conductance and PHS (Kennedy et al., 2019). Since the author's are already adjusting for SWC, adjusting parameters inherent to the soil hydraulics seems somewhat redundant.**

We agree that we could improve the simulation of ET by updating vegetation hydraulic parameters and will do so in further studies. However, for this particular study we chose to use parameter updating to improve the SWC estimation as best as possible and analyze the effect of this improvement to the related evapotranspiration estimation. This decision is related to a central objective of this study: To investigate whether precise in situ soil moisture measurements can improve ET characterization, as several studies have found that this cannot be achieved with more uncertain remote sensing soil moisture data. Please notice that assimilating soil moisture data with the aim to improve the simulation of land surface processes is a common modelling strategy.

Changes in text:

Lines: 227 - 230

"In this study, the parameters are chosen to optimize the SWC estimation and not ET estimation to study the effects of SWC improvements on ET. To more directly improve the ET estimation, parameters affecting the ET process directly should be added, e.g. vegetation hydraulic parameters."

**Finally, the authors never provide figures/tables of the influence of the DA on the parameter values. Do the parameter values converge to a new value, or are they temporally changing and random? Without this information it is impossible to assess why the parameter estimation had little influence on ET.**

We agree that we did not provide enough information about the parameter updates, since we did not see it as a focus of this study. The parameters updated with DA do change when observations are assimilated and converge to new values. We included a new figure to show this.

Changes in text:

Lines: 286 - 291

"Figure 5 shows time series of the estimated saturated soil hydraulic conductivity for each of the sites and the three observation layer depths. The DASP scenario results in parameter changes when the first observations are available but converge over the time of the simulation to a new value. The corresponding time series for the other soil hydraulic parameters can be found in the appendix (Figures A8, A9, and A10). The sand, clay, and organic matter fraction and thus the soil hydraulic parameters can vary with depth but as shown in Fig. 5 the DA updates to the parameters affect the different layers similarly."

[Figure]

Figure 5: Time series of the saturated soil hydraulic conductivity for each site in the DASP simulation. The grey line is the value at 5 cm depth, the blue line at 20 cm depth, and the green line at 50 cm depth.

**Detailed comments:**

**Line 55-60:  The literature review seems a bit incomplete.  Site level studies where parameters were hand tuned for the site of interest for forested sites also seems relevant:   Duarte at al, 2017; https://doi.org/10.5194/bg-14-4315-2017 and Raczka et al., 2016; https://doi.org/10.5194/bg-13-5183-2016.**

We focused in our literature review more on studies using data assimilation and agree that we can extend the literature review to include references to studies using manual tuning of parameters.

Changes in text:

Lines: 80 - 85

"Other studies have used manual tuning of parameters to improve CLM simulations for forests. For instance, Duarte et al. (2017) calibrated CLM4.5 for an old-growth coniferous forest and found good agreement between simulated and observed response of canopy conductance to atmospheric vapor pressure deficit and soil water content. Raczka et al. (2016) used CLM4.5 and implemented a seasonally varying calibration of vegetation parameters and accurately simulated net carbon exchange, latent heat exchange, and biomass."

**Line 61: crop cover, not cover crop**

The referenced study is about the inclusion of the management practice of cover crops, i.e. crops specifically planted to cover the soil during winter to reduce soil erosion, soil compaction, and nitrogen leaching and to increase agricultural productivity by nitrogen fixation. Therefore, we think the term "cover crop" is correct here.

**Line 69: "The point scale measurements use invasive equipment and the specific measurement volume, exact depth of the sensors, number of sensors, and number of stations varies from site to site. For a few sites we use soil water content measurements from Cosmic Ray Neutron Sensing (CRNS) from the COSMOS-Europe data set (Bogena et al., 2022)."**

**Seems worthwhile to diagnose the in-situ observations with the CRNS data for the same sites. This could help quantify representation of point level measurements for a flux tower.**

We agree that a study comparing the in-situ observation with the CRNS observations for sites where both data exist would be useful. However, in this study we focus on the data assimilation and the effect on modeled evapotranspiration and not the direct comparison between observational methods. This is already investigated and documented in other papers and goes beyond the scope of this study.

Changes in text:

Lines: 102 - 106

"CRNS use neutrons as proxy for SWC and the vertical measurement depth varies with the soil moisture conditions. Additionally, the uncertainty of CRNS-derived soil moisture varies not only with the different neutron detectors but also with the number of counts in a time period and therefore results under lower soil moisture conditions are more accurate (Bogena et al., 2022)."

**Line 73: "The CRNS provides continuous and non-invasive soil water content measurements over a spatial footprint of hundreds-of-meters and integrates from the surface to a depth of 10-70 cm vertically in the soil (Zreda et al., 2008; Köhli et al., 2015)."**

**If the COSMIC measurements are non-invasive, how are they integrating the soil water content from 10-70 cm vertical depth? Clearly these are not just measurements, but modeled output that is likely constrained from the surface measurements (1-5 cm). What are the uncertainty values for the CRNS values? More details are needed.**

The CRNS measures neutrons as proxy for SWC and use conversion functions and weighting to provide soil water content for the upper 10 cm of soil (under very wet conditions) to upper 70cm of soil (very dry conditions). The uncertainty of CRNS-derived soil moisture varies not only with the different neutron detectors but also with the number of counts in a time period and thus results under lower soil moisture conditions are more accurate.

Changes in text:

Lines: 102 - 106

"CRNS use neutrons as proxy for SWC and the vertical measurement depth varies with the soil moisture conditions. Additionally, the uncertainty of CRNS-derived soil moisture varies not only with the different neutron detectors but also with the number of counts in a time period and therefore results under lower soil moisture conditions are more accurate (Bogena et al., 2022)."

**Lines 65-83: Seems odd the authors would not mention their previous manuscript Strebel at al., (2022) in this review section (Wustebach catchment), which essentially uses the same exact DA setup, but expands the analysis to more sites in this analysis. Seems the authors should state their previous findings here, and use that to motivate and distinguish this analysis from the previous one.**

The previous study mentioned here focused more on the implementation details of the software coupling and therefore we reference it in the method section.

Changes in text:

Lines: 89 - 93

"In a previous study (Strebel et al. 2022), we investigated the potential for data assimilation of in-situ SWC measurements to improve model estimation for a single forest site. This study expands this method to more forest sites and investigates the effect of improved SWC estimation on ET."

**Line 97: 'daily average soil water content data are assimilated'. Need more detail of what depth these observations are taken from. This needs to be explained either here or more clearly in the methods.**

We describe the vertical layout in section 2.4.1.

Changes in text:

Lines: 187 - 191

"For FLUXNET sites, measured soil water content is provided for up to three depths described as superficial, medium, and deep. Since data assimilation in CLM5-PDAF requires a specific vertical layer, we assigned 5, 20 and 50 cm to the respective FLUXNET SWC layers. For the CRNS sites, the measurement depth for each individual measurement is calculated following Schrön et al. (2017) and is included in the dataset from Bogena et al. (2022). "

**Table 1: I think elevation would also be relevant to report here.**

Added elevation to table 1.

Changes in text:

**Table 1**: Overview of the study sites. Classification uses the International Geosphere-Biosphere Program Code (IGBP) as is used for FLUXNET: MF for mixed forests, ENF for evergreen needle leaf forests, DBF for deciduous broad leaf forests, EBF for evergreen broad leaf forests, WSA for woody savannah. LON is longitude and LAT latitude.

| Site name | Country | Abbreviation | Code | LON | LAT | Elevation [m.a.s.l.] | Data source | Mean annual temperature [°C] | Mean annual precipitation [mm] | Typical tree species |
|---|---|---|---|---|---|---|---|---|---|---|
| Brasschaat | Belgium | BE-Bra | MF | 4.51 | 51.3 | 16 | FLUXNET | 9.8 | 750 | Scots pine |
| Bílý Kříž forest | Czech Republic | CZ-BK | ENF | 18.53 | 49.5 | 875 | FLUXNET | 7 | 1316 | Norway spruce |
| Hainich | Germany | DE-Hai | DBF | 10.45 | 51.07 | 430 | FLUXNET | 8.3 | 720 | Mixed Beech |
| Hohes Holz | Germany | DE-HoH | DBF | 11.21 | 52.08 | 217 | COSMOS Europe | 10 | 820 | Mixed beech |
| Oberbärenburg | Germany | DE-Obe | ENF | 13.72 | 50.78 | 734 | FLUXNET | 5.5 | 996 | Norway spruce |
| Wüstebach | Germany | DE-Wue | ENF | 6.33 | 50.5 | 605 | COSMOS Europe | 7 | 1180 | Spruce |
| Gludsted | Denmark | DK-Glu | ENF | 9.33 | 56.07 | 86 | COSMOS Europe | 8.2 | 1080 | Spruce |
| Conde | Spain | ES-Cnd | WSA | -3.22 | 37.91 | 370 | FLUXNET | 15.8 | 474 | Olive grove |
| Hyytiälä | Finland | FI-Hyy | ENF | 24.29 | 61.84 | 181 | LTER Europe | 3.8 | 709 | Boreal Scots pine |
| Sodankylä | Finland | FI-Sod | ENF | 26.63 | 67.36 | 180 | FLUXNET | -1 | 500 | Boreal Scots pine |
| Puéchabon | France | FR-Pue | EBF | 3.59 | 43.69 | 270 | FLUXNET | 13.5 | 883 | Evergreen oak |
| Lavarone | Italy | IT-Lav | ENF | 11.28 | 45.95 | 1353 | FLUXNET | 7.8 | 1291 | Coniferous forest |
| Loobos | Netherlands | NL-Loo | ENF | 5.74 | 52.16 | 25 | FLUXNET | 9.8 | 786 | Scots pine |

**Figure 1: Would also be helpful to include information about data source in this figure too, similar to Table 1.**

We agree and we have included the information the legend of Figure 1.

**Line 116: 'partitions' is strange in this context. Suggest 'CLM5 simulates sensible and latent heat flux for both vegetated and ground fluxes'**

We agree.

Changes in text:

Lines: 154 - 155

"CLM5 simulates sensible and latent heat flux for both vegetated and ground fluxes."

**Line 120: same comment as above, suggest: 'canopy evaporation is represented as the sum of steam and leaf evaporation as a function of temperature.'**

We agree.

Changes in text:

Lines: 158 - 160

"Interception, throughfall and canopy drip are explicitly modeled in CLM5 and canopy evaporation is represented as from the sum of stem and leaf surface evaporation as a function of temperature."

**Line 127: How many ensemble members were used to sample the model uncertainty?**

We used an ensemble of size 96 for all our simulations.

Changes in text:

Lines: 174 - 175

"In this study we use an ensemble of 96 member to sample the model uncertainty."

**Line 144: "For simulations assimilating CRNS, H assigns the mean observed SWC to all the layers down to the measurement depth."**

**It is unclear what this means. Guessing from previous statements that CRNS is the integrated water content from 10-70 cm, so are you comparing against the CLM modeled soil layers that**

**coincide with 10-70 cm?   You need to be more explicit of how you calculate your observation operator in general.  What CLM soil layer variables are you using to calculate the expected observation?   Also for the point measurement of soil water content, where are the depth measurements?  I don't see this mentioned in this section where it would be appropriate.**

**I see you describe the FLUXNET soil water content in lines 176 through 179, but it seems out of place.  This description should come earlier and discussion of the observation operator should be in one place.   The statement 'For the CRNS sites, the measurement depth for each individual measurement is calculated following Schrön et al. (2017) and is included in the dataset from Bogena et al. (2022).' Seems to conflict with an earlier statement that the CRNS integrates between 10-70 cm. depth.**

For CRNS the measurement depth varies as function of soil water content. The cited "10-70" cm is the usual range of the measurement depth. Since the reference data set provides the measurement depth for each observation the depth which is used by H also varies.

Changes in text:

Lines: 187 - 191

"For FLUXNET sites, measured soil water content is provided for up to three depths described as superficial, medium, and deep. Since data assimilation in CLM5-PDAF requires a specific vertical layer, we assigned 5, 20 and 50 cm to the respective FLUXNET SWC layers. For the CRNS sites, the measurement depth for each individual measurement is calculated following Schrön et al. (2017) and is included in the dataset from Bogena et al. (2022). "

**Line 159:  "By default, CLM5-PDAF updates soil hydraulic parameters through changes to fractions of sand, clay, and organic matter and the pedotransfer function of Clapp and Hornberger (1978)."**

**This statement and methodology is problematic.  See my comment in summary section above.**

See our comment above.

**Line 162:  It seems you are allowing the % sand, clay and organic to vary by layer?   Does this make physical sense?  This would allow for each layer to vary independently with no fixed bounds other than 0 or 100%?   I cannot find any results that show the adjusted parameter values.**

The sand, clay, and organic matter fraction already vary by layer even in the default CLM5 surface files. This makes physical sense, especially for organic matter that is usually very high in the first few layers and rapidly decreases for any deeper layers. During parameter updates the layers are updated using the Kalman gain, which ensures that spatial correlation in soil texture between soil layers is taking into account.

Changes in text:

Lines: 221 - 227

"We tested directly updating saturated hydraulic conductivity, porosity, hydraulic conductivity exponent, and soil matric potential but this resulted in more unstable estimates than indirectly updating soil hydraulic parameters. The pedotransfer function which is used for the indirect updating results in reasonably correlated soil hydraulic parameters. "In testing a direct approach to updating saturated hydraulic conductivity, porosity, hydraulic conductivity exponent B, and soil matric potential we found that updating the parameters indirectly to provide more stable simulations. The pedotransfer function keeps the soil hydraulic parameters reasonably correlated to each other."

**Line 174: You are perturbing the % sand, clay and organic to generate ensemble spread, but you are also estimating these parameters at the same time. How do you maintain ensemble spread, or prevent the system from collapsing on a solution?**

**It seems you address this in Line 192, but seems out of place to mention it here. Should come earlier.**

Yes, the sand, clay, and organic matter fractions are all perturbed to create the ensemble spread and updated. However, the atmospheric forcing variables are also perturbed and create ensemble spread and prevent a collapsing on a solution.

Changes in text:

Lines: 218 - 219

"A damping factor of 0.1 is used on the parameter updates to avoid filter inbreeding and keep the ensemble spread larger so that the model error covariance matrix is a good approximation for model uncertainty."

**Line 226: "This indicates that the SWC-ET relation is incorrect for these sites. A possible explanation is the water uptake by roots in the deeper layers is underestimated for forest sites, as also suggested by Shrestha et al. (2018)."**

**I don't see how you can come to this conclusion without diagnosing your leaf area (LAI) mismatch. See my general comment/concern above.**

We rephrased the statement.

Change in text:

Lines: 311 - 313

"Another possible explanation for the improvement in SWC estimation but no improvement of ET estimation is the underestimation of root water uptake from deeper soil layers for forest sites, as also suggested by Shrestha et al. (2018)."

**Line 315-325: Exactly. Prescribing LAI through CLM5-SP or assimilating LAI observations is necessary to demonstrate that soil water content does not improve ET, and that it is a parameter problem.**

We agree that assimilating LAI will improve ET. However, we still think that it is noteworthy that data assimilation of high-quality in-situ soil water content provides no improvement to ET estimation for forested sites with CLM5-BGC.

**Line 337: "These results suggest that state-of-the-art LSM such as CLM5 still suffer from uncertainties in the representation of soil hydrological processes in forests, e.g. deep root water uptake, uncertainties in the representation of biological processes of tree transpiration, partly related to uncertain vegetation parameters."**

**I disagree, at least I don't think your results demonstrate this. The inability to prescribe leaf area, or account for the assimilation of leaf area (Fox et al., 2022) does not allow you to make this statement with certainty.**

We rephrased the statement to account for the lack of LAI adjustment in this study.

Changes in text:

Lines: 451 - 456

"These results suggest that improving the SWC estimation of state-of-the-art LSM such as CLM5 is not sufficient to improve ET estimation for forest sites. To improve ET estimation it is also necessary to consider the representation of LAI in magnitude and timing, as well as uncertainties in water uptake by roots and vegetation parameters."

**Furthermore the parameter estimation performed here does not seem to directly update 'true' vegetation hydraulic parameters (e.g. vegetation hydraulic parameters; Kennedy et al., 2019) to further test this theory. You adjust soil hydraulic parameters which could influence soil moisture dynamics, but seems redundant in that soil moisture is already adjusted directly.**

**It also would have been useful to include the prior and posterior values for your parameter estimation. Its unclear if the lack of impact the parameter estimation had on ET, was because of lack of sensitivity to controlling the ET, or because the parameters were not updated significantly from their prior values.**

Yes, we did not take vegetation hydraulic parameters into account in this study as we only update soil hydraulic parameters which is not necessarily redundant but usually provides increased improvement in SWC estimation. However, we agree that it is not enough for ET estimation and we will take vegetation parameters and LAI assimilation into account in our future studies.

Changes in text:

Lines: 227 - 230

"In this study, the parameters are chosen to optimize the SWC estimation and not ET estimation to study the effects of SWC improvements on ET. To more directly improve the ET estimation, parameters   affecting the ET process directly should be added, e.g. vegetation hydraulic parameters."

**Figures 2-4:  You could combine these plots to present a better comparison between OL-DAS and OL-DASP.**

We combined the Figures 2-4.

Changes in text:

Lines: 272 - 281

"Figures 2 and 3 show the results of the soil water content simulations at 20 cm depth of the OL, DAS and DASP simulations compared to the soil water content observed at the nine sites. Figure 2 compares the OL and DAS results and Fig. 3 compares the OL and DASP results. The corresponding scatter diagrams for the depths 5, 20, and 50 cm can be found in the appendix (Figures A1 - A7). Overall, the results show expected improvements by data assimilation of observed soil water content. For the OL simulations, Fig. 2 shows particularly large RMSE values for CZ-BK, DE-Obe, FI-Sod and NL-Loo. Fig. 2 also illustrates the improved performance achieved by DASP, with a RMSE reduction from 29.3 cm³/cm³ to 6.25 cm³/cm³ and a MBE reduction from 28.06 cm³/cm³ to -2.94 cm³/cm³ for FI-Sod. Parameter updating, as shown in Fig. 3, further improves the simulation results, but the improvement from DAS to DASP is significantly less than from OL to DAS."

[Figure]

**Figure 2: Scatter plots of observed soil water content at 20 cm depth at nine study sites versus OL and DAS simulated soil water content. The points represent daily averages for the days observation data are available. Green points are OL and blue points are DAS results.**

[Figure]

**Figure 3: Scatter plots of observed soil water content at 20 cm depth at nine study sites versus OL and DASP simulated soil water content results at 20 cm depth. The points represent daily averages for the days observation data are available. Green points are OL and purple points are DASP results.**

**Figure 11:   This figure seems very critical, given the potential mismatch between observed and simulated LAI, and the potential impact that has on the soil water content vs. ET relationship.  I feel like you also need to present the average season cycle of LAI for all these sites, because the the annual average is not enough.  CLM has trouble simulating the timing of seasonal phenology of LAI especially for evergreen forested sites (not just magnitude). Would help to simulate a CLM5-BGC and CLM5-SP simulation here, where LAI is prescribed, or assimilate LAI observations directly for your experiment.**

We added a figure with the seasonal LAI cycle for each site and include a comparison with simulations where LAI is prescribed as mentioned earlier. In addition, we will work with LAI assimilation in future studies.

Changes in text:

Lines: 303 – 306, 354 - 355

"This could be caused by the mismatch of simulated and actual LAI for these sites. To investigate this, we repeated the simulations using CLM5 with satellite-derived phenology (CLM5-SP) The results are shown in Fig. 9. For CLM5-SP we observe an average improvement in the RMSE of SWC between 57.6 % and 64.3 % and an average RMSE deterioration of 5.8 % for the ET estimation. These CLM5-SP simulations use the default datasets from CLM5 and without site specific calibration of the timing or magnitude of the seasonal phenology of LAI. Therefore, even for the CLM5-SP simulations, there is a mismatch between simulated and actual LAI. However, also for this case, there are sites with a large improvement in SWC estimation that show deterioration in the ET estimation."

"Figure 14 shows the LAI for each site averaged over all the simulated years. Sites with the same PFT show clear differences in the yearly LAI cycle. "

[Figure]

**Figure 14: Seasonality of simulated LAI (DASP) for each of the sites.**

**Figure 12:  These plots could be useful to diagnose the behavior, but I am afraid information is 'washed out' when averaging over all years and all sites.   This brings up some really important questions, such as, did the assimilation adjust the SWC all the way through the**

**root zone, or was the adjustment primarily superficial, and limited to the upper 25-50 cm. only?   Is the majority of the root zone below this, and contribute to the lack of impact on ET?**

We show results averaged over all years and sites to give an overview on general trends in the results. However, we agree that additional figures can shed a further light on the results. The data assimilation adjusts the SWC for all layers, but as shown in this figure the change is usually larger in the upper layers than in deeper layers.

Changes in text:

Lines: 341 - 343

"Although DASP adjusts SWC at 5cm towards the observations, the correction for SWC at 50cm depth is smaller because not all sites provide data at this depth. However, for all sites the data assimilation provides some improvement for SWC estimation even in layers below the observation depth."

---

## Author Response (AR2)

**Scientific/Detailed Comments: Referee comments in bold and author answer non-bold.**

**This reviewer appreciates the author's responses to the initial set of questions and concerns. This reviewer also appreciates the authors running additional simulations using the CLM5-SP implementation (prescribing LAI) to help better diagnose why adjusting of SWC does not impact ET. This reviewer believes this work is potentially relevant and useful to the scientific community, and acknowledges the considerable amount of work and preparation to performing multiple site level runs. However, this reviewer still has multiple critical concerns that were not addressed by the manuscript revisions:**

**1) Although the authors implement CLM5-SP simulations to prescribe LAI, it seems they have re-used the same optimized parameters (sand/clay/organic fraction) based on the CLM5-BGC simulations. These parameters are inappropriate to use with CLM5-SP, because they have compensated for biases in LAI during the CLM5-BGC runs and would not allow for a proper diagnosis of the impact of LAI on ET. The author's should perform a separate parameter optimization with CLM5-SP.**

We thank the reviewer for the comment, but there has been a misunderstanding. We did not re-use the same optimized parameters based on CLM5-BGC as assumed by the reviewer. For both CLM5-SP and CLM5-BGC simulations, we start with the same values for sand, clay, organic fraction based on the default CLM5 surface file creation data sets. For the data assimilation with parameter updates the method, as described in section 2.3.2, is used for both CLM5-BGC and CLM5-SP simulations which results in different parameter updates for the CLM5-BGC and CLM5-SP simulations. Therefore, the CLM5-SP simulations do not contain any information from the CLM5-BGC simulations.

To clarify this, we added the following sentence: "The CLM5-SP OL and DASP simulations do not use any information from the CLM5-BGC simulations which implies that for the CLM-SP DASP simulations parameters are estimated independently from the CLM5-BGC simulations".

Lines: 320 - 323

**2) The author's have not provided a full SWC profile (1-100 cm) of the prior and posterior values to compare against the root profile for each site. Without this information, it is unclear if the update of the SWC signficantly impacts soil moisture below 5 cm. The author's only provide 5cm and 50 cm snapshots of all-site averages of these values, which makes it impossible to diagnose why SWC updates have minimal impact on ET. It could be the SWC updates are only superficial (~5 cm) and do not overlap with the majority of the root profile (1-100 cm). The authors do not present the reader with a profile of the root profile so again, diagnosis of the results remain challenging.**

We thank the reviewer for this suggestion. We have added figures showing the SWC profiles for OL and DASP and the root profile for each site. Figure 5 shows how SWC is updated throughout the soil layers for different sites and as function of depth.

[Figure]

**Figure 5**: Profile plots for the first 10 layers, showing the root fraction and the time-averaged SWC per depth for each site. In the SWC profiles, the red and green lines represent the SWC from the open-loop simulations (OL) and DASP simulations, respectively.

Changes in text:

Figure 5 shows the vertical profile of the root fraction and the ensemble SWC average for the OL and DASP simulations at each site. The SWC is updated for all layers, including the layers with the largest root fraction, but the magnitude of the changes varies strongly between sites. For most sites the data assimilation changes the absolute SWC values while the shape of the vertical profile remains similar to the OL simulations. FI-Hyy and FI-Sod show a different response with SWC decreases in the upper 25 to 50 cm and a SWC increase in the deeper layers.

Lines: 291 - 296

**3) This reviewer still has concerns about the methodology to update soil/clay/ organic matter fraction to indirectly influence the true soil hydraulic parameters. This reviewer sympathizes that when the authors attempted to update the soil hydraulic parameters directly they experienced edge-hitting conditions. However, this was likely because these parameters were compensating for other significant model structural error in the original simulations (biased LAI). Would recommend that the authors optimize the true soil hydraulic parameters using CLM5-SP.**

We understand the concerns of the reviewer, but we like to emphasize that the focus of this study is on CLM5-BGC. We included some results from CLM5-SP only for comparison as recommended by the reviewer. We therefore believe that these suggestions go beyond the scope of this work. In addition, as already mentioned before, the indirect updating of soil hydraulic parameters through the use of sand/clay/organic matter fractions and pedotransfer function is an established method. This has also been shown in many studies that we have cited in both the introduction chapter and in our response in the first revision.

**4) Putting this reviewer's objections to the methodology aside (point 3), the author's still have not provided the prior and posterior values of the soil/clay/organic parameters that they optimized during this study. Therefore it is impossible to diagnose how significant the parameter updates were, if the updates were physical, or if the updates occurred across the full root profile (1-100 cm). This makes it impossible to address to what extent these parameter updates should impact ET.**

We thank the reviewer for this suggestion. In the previous revision, we already included Figure 5, A8, A9, A10 to show the time evolution of the hydraulic parameters. In this revision, we have replaced the previous Figure 5 with Figure 7, 8, and 9 for the prior / posterior profile of the soil/clay/organic parameters.

[Figure]

**Figure 7**: Profile plot showing the sand fractions for the first 10 layers of all 13 sites.

[Figure]

**Figure 8:** Profile plot showing the clay fractions for the first 10 layers of all 13 sites.

[Figure]

**Figure 9**: Profile plot showing the organic matter fractions for the first 10 layers of all 13 sites.

**This reviewer commends the authors for including CLM5-SP simulations which is a step in the right direction. However, for all of thes reasons stated above, regretfully, this reviewer cannot recommend this manuscript for publication. See detailed responses to the author's responses to tracked changes version of the manuscript below:**

**Response to Review comments by Referee #2:**

**"This could be caused by the mismatch of simulated and actual LAI for these sites. To investigate this, we repeated the simulations using CLM5 with satellite-derived phenology (CLM5-SP). The results are shown in Fig. 9. For CLM5-SP we observe an average improvement in the RMSE of SWC between 57.6 % and 64.3 % and an average RMSE deterioration of 5.8 % for the ET estimation."**

**To first order – before even looking at SWC and ET estimation, did the CLM5-SP simulations improve the simulated LAI as compared to the CLM5-BGC simulations? Where do you show this? I was hoping you would have updated Figure 11 and Figure 12 and Figure 14 with your improved LAI for the CLM5-SP runs, but you haven't done this. In what way is LAI impacting the SWC? It's improving the RMSE SWC, but why? Is it improving the biases, or is the seasonal pattern improved?**
* * *
We thank the reviewer for this suggestion. We have now included the LAI comparison for CLM5-BGC and CLM5-SP in Fig 18. However, since the focus of the study is not to compare these two different models but to study the impact of SWC assimilation on the ET estimation of CLM5-BGC, we believe that investigating the difference between CLM5-SP and CLM5-BGC to this extent is beyond the scope of this work. As already mentioned in a previous response, we will take up this topic in a future study.

Changes in text:

"Figure 18 shows the LAI for each site averaged over all the simulated years and the difference between the prescribed LAI used in CLM5-SP and the estimated LAI from CLM5-BGC. Sites with the same PFT show clear differences in the yearly LAI cycle."

Figure 18 (formerly 14) has been changed.

[Figure]

Figure 18: Seasonality of simulated LAI for each of the sites. The red line represents predicted LAI from the CLM5-BGC-DASP simulations and the blue line represents LAI inputs used in the CLM5-SP-DASP simulations.

Lines: 371 - 373

"These CLM5-SP simulations use the default datasets from CLM5 and without site specific calibration of the timing or magnitude of the seasonal phenology of LAI. Therefore, even for the CLM5-SP simulations, there is a mismatch between simulated and actual LAI."

Yes, the default approach in CLM5-SP is for the site level LAI to be prescribed by the site level climatology, however, the user can also prescribe the site level LAI manually from the site level observations. This would be the recommended approach for this application. Because the author's identify there is a mismatch between simulated and actual LAI, they *have* the observed LAI so it is not clear why they did not use it for the CLM5-SP simulations.
* * *
We agree that for CLM5-SP studies the best approach would be to prescribe the site level LAI manually, however we do not have the observed LAI for these sites at hand. Unfortunately collecting and preparing the site level LAI for a comparison with CLM5-SP is beyond the scope of this work. We will take up this topic in a future study, as already mentioned in a previous response.

Changes in text:

"Because the focus of this study is on CLM5-BGC, these CLM5-SP simulations use the default datasets from CLM5 since in-situ LAI measurements for these sites were not available. "

Lines: 319 - 320

**"Another possible explanation for the improvement in SWC estimation but no improvement of ET estimation is the underestimation of root water uptake from deeper soil layers for forest sites, as also suggested by Shrestha et al. (2018). "**

**Here are other more likely explanations for the lack of improvement of ET. First, the authors have seemed to re-use the same optimized parameters from their previous CLM5-BGC DA and DASP simulations and applied those same parameters to the CLM5-SP simulations. If it was done that way, the methods are incorrect. The optimized parameters from the CLM5-BGC simulations are biased because they compensated for strong biases (error) in the LAI. The CLM5-SP simulations have removed the majority of the simulated LAI bias, and therefore the DA and DASP simulations should have been run again to calibrate \*new\* parameters with CLM5-SP.**

**A second reason is that the authors have not convincingly shown that updates to the SWC or soil/clay/organic parameters were significant, or coincided with the majority of the root profile, which is the critical factor in improving the water root uptake and ET.**

**A third reason for the disconnect between SWC and ET is that the authors optimized clay/sand/organic soil fraction and parameters related to root uptake of soil water, but \*did not\* look at hydraulic parameters related to the transfer of water from root to leaf, or parameters related to stomatal conductance which are critical for controlling ET. This reviewer recommended these authors take this approach in the previous set of reviews and this reviewer's advice was disregarded.**
* * *
We thank the reviewer for the comment, but we did not re-use the same optimized parameters as explained in the answer above. We have now included a figure showing the SWC update in profile and the root fraction profile. As mentioned before, the parameters chosen for updating are designed to improve the SWC representation of the model. The objective of this study is to improve model SWC estimation with data assimilation of high-quality in-situ observations and updating of the SWC relevant parameters and then see how this improved SWC impacts the ET estimation. Nevertheless, we agree that this topic is relevant and mention in both the discussion and conclusion that adding additional ET specific parameters are necessary to further improve ET estimation. However, since this goes beyond the scope of this work, we will take up this topic in a future study.

Changes in text:

"In this study, the parameters are chosen to optimize the SWC estimation and not ET estimation to study the effects of SWC improvements on ET. To more directly improve the ET estimation, parameters that are critical to the ET process should be added, e.g. vegetation hydraulic parameters that are related to the transfer of water from the root to leaf or parameters related to stomatal conductance."

Lines: 231 - 235

**"Soil texture data are always subject to uncertainties and therefore it is reasonable to not fix them. Therefore, the indirect approach of updating soil characteristics, i.e. sand, clay, and organic matter fraction, to update the soil hydraulic parameters using a pedotransfer function was used in various studies (Naz et al., 2019; Han et al., 2014; Baatz et al. 2017)."**

**Treating soil clay and organic matter fractions as parameters to be calibrated, to indirectly improve the soil conductivity of water, instead of directly calibrating the plant and soil hydraulic parameters directly, is not justified in this reviewer's opinion. This would be similar to trying to calibrate a photosynthetic parameter to improve the simulation of photosynthesis, but instead of calibrating the photosynthetic parameter directly, calibrating the % plant functional type distribution at the site.**
* * *
We thank the reviewer for the comment, but we disagree that the indirect approach is not justified and as shown in the citations has been used as a method in previous studies. The indirect approach is more similar to combining the parameters that are related to each other with a well established function (in our case the pedotransfer function from Clapp and Hornberger (1978)) and using the combined proxy parameter in the update instead of calibrating multiple parameters independently.

Changes in text: See previous answer.

**"Secondly, we performed data assimilation using a direct approach to update combinations of soil saturated hydraulic conductivity, hydraulic conductivity exponent, porosity, and saturated soil matric potential but found that indirectly updating sand, clay, and organic matter fractions using pedotransfer functions provides better results. Specifically, we found that parameters often approached upper or lower defined limits, which did not happen with the indirect approach. Updating the solid fractions provided in general more stable parameter estimates and better simulation results."**

**What are you going to learn by optimizing sand/clay and organic matter fractions, without validating them against observations? What can you learn by this to improve other simulations? I don't see how that is advancing the scientific understanding of the relationship between soil moisture and ET. It is more likely you just haven't optimized the parameters that are most influential in the control between ET and soil moisture – those related to stomatal conductance and hydraulic parameters that control the water transfer from root to leaf.**
* * *
We thank the reviewer for the comment. Firstly, the purpose of the parameter updating with the state augmentation approach is to improve the SWC estimation during the simulation based on the

observations. Secondly, the indirect approach is helpful to make use of the optimized parameters in further simulations. Because the updated sand, clay, and organic matter fraction can be easily added to the surface properties while currently CLM5 does not have the soil hydraulic parameters in any input file and therefore requires code changes to include in any further simulations for the same sites. Finally, as mentioned previously we agree that other parameters are also important to the SWC – ET relationship and we will include them in further studies.

Changes in text:
"In this study, the parameters are chosen to optimize the SWC estimation and not ET estimation to study the effects of SWC improvements on ET. To more directly improve the ET estimation, parameters that are critical to the ET process should be added, e.g. vegetation hydraulic parameters that are related to the transfer of water from the root to leaf or parameters related to stomatal conductance."

Lines: 231 - 235

**"Reviewers first comment:**
**Furthermore, the choice of parameters to be estimated should more closely relate to processes controlling the SWC-ET relationship, such as stomatal conductance, and the \*vegetation\* hydraulic parameters specifically involved in the Plant Hydraulic Stress formulation in CLM5. The soil hydraulic parameters estimated in the manuscript, seem to control the dynamics of moisture within the soil layer, but not the transfer of water from root to leaf, which is controlled by the stomatal conductance and PHS (Kennedy et al., 2019). Since the author's are already adjusting for SWC, adjusting parameters inherent to the soil hydraulics seems somewhat redundant.**

**Author's Response:**
**We agree that we could improve the simulation of ET by updating vegetation hydraulic parameters and will do so in further studies. However, for this particular study we chose to use parameter updating to improve the SWC estimation as best as possible and analyze the effect of this improvement to the related evapotranspiration estimation. This decision is related to a central objective of this study: To investigate whether precise in situ soil moisture measurements can improve ET characterization, as several studies have found that this cannot be achieved with more uncertain remote sensing soil moisture data. Please notice that assimilating soil moisture data with the aim to improve the simulation of land surface processes is a common modelling strategy."**

**This reviewer is not satisfied with this explanation. In the author's previous site level SWC calibration study (Strebel et al., 2022, GMD) these authors stated the following in the published version of their Discussion and Conclusions:**

**"We were not able to show significant impact of the assimilated soil water content on the evapotranspiration flux. In a future study, we will investigate whether this is the case for other study sites and in other climates. We will also include other variables and parameters in the data assimilation to test their effects on the evapotranspiration flux.**
**The performance of CLM5-PDAF could be further improved by updating soil hydraulic parameters themselves, instead of indirectly updating them via soil texture and pedotransfer functions. This could potentially reduce the model uncertainty further since the accuracy of the pedotransfer functions would be less of an issue after parameter updating. This will require more fundamental code changes and will be considered in future work. In addition,**

**CLM5-PDAF will be further extended by the assimilation of more state variables, like for example LAI or soil temperature.”**

**However, contrary to their own recommendations, this manuscript essentially takes the same approach as the Strebel at al., 2022 manuscript, except applies it to more sites, and comes to the same conclusion: essentially optimizing the SWC has very little benefit to ET. The author's need to do a better job of defending what added benefit this brings to the scientific community beyond the original study. As far as I can tell the 'new' addition to this manuscript is the use of CLM5-SP to prescribe LAI (as recommended by this reviewer), to demonstrate to what extent the prescription of LAI improves SWC and ET, however, the author's methods of simulating CLM5-SP seem to be seriously flawed, and the results are not sufficiently included/discussed in the revised manuscript.**
* * *
Yes, we specifically mention that this study takes the same approach as Strebel et al. 2022 and applies it to more sites. Strebel et al. 2022 focuses on the implementation details of the CLM5-PDAF coupling and uses only one site to highlight the results and therefore the conclusions there only applies to that site. The benefit of this study is that it covers sites throughout Europe from various climates, various PFTs, and different measurement sources for the in-situ SWC observations. Only this coverage allows more general conclusions to be drawn. We have added the LAI inputs used in CLM5-SP to the figure showing CLM5-BGC LAI. In addition, we implemented the direct updating of soil hydraulic parameters in the code and tested this, but found that this approach yielded worse results than indirect parameter updating.

Changes in text:
"This study expands this method to more forest sites and investigates the effect of improved SWC estimation on ET. Investigating the method for a large number of sites is the important contribution of this study and necessary to show that the conclusions from Strebel et al. (2022) are not just a characteristic of the one study site but apply more broadly to forest sites simulated with CLM5."

Lines: 91 - 95

**Reviewer's original comment:**
**Line 162: It seems you are allowing the % sand, clay and organic to vary by layer? Does this make physical sense? This would allow for each layer to vary independently with no fixed bounds other than 0 or 100%? I cannot find any results that show the adjusted parameter values.**

**Author's Response:**
**The sand, clay, and organic matter fraction already vary by layer even in the default CLM5 surface files. This makes physical sense, especially for organic matter that is usually very high in the first few layers and rapidly decreases for any deeper layers. During parameter updates the layers are updated using the Kalman gain, which ensures that spatial correlation in soil texture between soil layers is taking into account.**

**Reviewer response:**

**OK – so where are the prior and posterior values of sand/clay and organic matter fraction? These are the actual values that you have updated in the DASP simulation, and these are still not included within the manuscript. The reviewer has no way to interpret how the assimilation influenced the sand/clay and organic matter fraction. Are these values**

**reasonable, physical, does the posterior value reach a stable value or does it vary over time (which would be unphysical). The author's have not provided this requested data.**
* * *
We thank the reviewer for the suggestion. We now included the evolution of the soil hydraulic parameters over the simulation time as well as the prior and posterior of the sand-, clay-, and organic matter fractions.

Changes in text:
"Figures 7, 8, and 9 show the initial (prior) and the updated (posterior) vertical profiles for the sand-, clay-, and organic matter fractions for the upper 1.2 meters (10 soil layers). The updated parameters often keep the profile distribution but have reduced or increased values throughout the layers compared to the prior."

Figures 7,8, and 9 are new, see answer above.

Lines: 302 - 304

**Reviewer original response:**
**Line 337: It also would have been useful to include the prior and posterior values for your parameter estimation. Its unclear if the lack of impact the parameter estimation had on ET, was because of lack of sensitivity to controlling the ET, or because the parameters were not updated significantly from their prior values.**

**Author response: Yes, we did not take vegetation hydraulic parameters into account in this study as we only update soil hydraulic parameters which is not necessarily redundant but usually provides increased improvement in SWC estimation. However, we agree that it is not enough for ET estimation and we will take vegetation parameters and LAI assimilation into account in our future studies.**

**Reviewer response: The author's response does not address the reviewer's question. The author's have not included the prior and posterior values of the clay/sand and organic matter fraction thus there is no way to address how the parameter adjustment is impacting SWC and ET.**
* * *
We now also include the prior and posterior of the sand-, clay-, and organic matter fraction profiles.

Changes in text:
"Figures 7, 8, and 9 show the initial (prior) and the updated (posterior) vertical profiles for the sand-, clay-, and organic matter fractions for the upper 1.2 meters (10 soil layers). The updated parameters often keep the profile distribution but have reduced or increased values throughout the layers compared to the prior."

Figures 7,8, and 9 are new, see previous answer.

Lines: 302 - 304

**"We added a figure with the seasonal LAI cycle for each site and include a comparison with simulations where LAI is prescribed as mentioned earlier. In addition, we will work with LAI assimilation in future studies."**

**There is a Figure 14 within the document with seasonal LAI plots, but it is unclear what simulation this is related to? It says the DASP simulation, but presumably this is with CLM5-BGC. How does the LAI compare between the CLM5-BGC and the CLM5-SP, and the site level observed LAI? LAI is a strong control on ET, thus more discussion needs to be brought upon the impact of LAI on ET. The authors briefly say that prescribing LAI shows strong improvement in RMSE of SWC, but deprecated improvement in ET. But it seems the CLM5-SP simulation was using DASP optimized parameters from the CLM5-BGC simulation. These optimized parameters are not compatible with CLM5-SP.**
* * *
We thank the reviewer for the comment. In the previous revision, we added Figure 14 to show the CLM5-BGC LAI. In this revision, we added LAI values of CLM5-SP to the same figure. We do not have site level LAI data to compare the model results to observations (see previous comment) and we did not use any parameters from CLM5-BGC for CLM5-SP.

Changes in text:

"Figure 18 shows the LAI for each site averaged over all the simulated years and the difference between the prescribed LAI used in CLM5-SP and the simulated LAI by CLM5-BGC. Sites with the same PFT show clear differences in the yearly LAI cycle."

Figure 18 (formerly 14) has been updated.

Lines: 371 - 374

**Reviewer original questions:**
**Figure 12: These plots could be useful to diagnose the behavior, but I am afraid information is 'washed out' when averaging over all years and all sites. This brings up some really important questions, such as, did the assimilation adjust the SWC all the way through the root zone, or was the adjustment primarily superficial, and limited to the upper 25-50 cm. only? Is the majority of the root zone below this, and contribute to the lack of impact on ET?**

**Author response:**
**"Although DASP adjusts SWC at 5cm towards the observations, the correction for SWC at 50cm depth is smaller because not all sites provide data at this depth. However, for all sites the data assimilation provides some improvement for SWC estimation even in layers below the observation depth."**

**Reviewer response: For the adjustment in SWC to have an important impact on the assimilation and especially ET, the SWC adjustment must occur across the root profile. The author's show significant SWC adjustment at 5 cm, and negligible adjustment at 50 cm. However, the root profile for these sites likely extends to 100 cm and below (the author's do not provide the root profile distribution for their sites, which can be output from CLM, thus the reviewer has no way to diagnose this). The authors have not convincingly shown that their SWC adjustment has made a significant impact across the full root profile. They should provide a figure of the full 1-100 cm SWC profile. Also averaging across all sites 'washes out' the site specific information which could allow for a proper diagnosis of how SWC adjustment impacts the root profile (1-100cm and below) for each site.**
* * *
We have added a figure to show the SWC update throughout the profile and the root fraction profile.

Changes in text:
"Figure 5 shows the depth profile for the root fraction and the SWC average of the OL and DASP simulations for the first 1.2 meters (10 layers)  for each site. The SWC is updated for all layers, including the layers with the largest root fraction, but depending on the site the magnitude of the update varies with depth. For most sites the data assimilation shifts the SWC values while keeping the profile similar to the OL results. FI-Hyy and FI-Sod are the exception and show a decrease of SWC in the first 25 to 50 cm and an increase of SWC in the deeper layers for DASP."

Figure 5 is new, see previous answer.

Lines: 291 - 296